# Learning Robust Representations with Long-Term Information for Generalization in Visual Reinforcement Learning

**Rui Yang**[1,2]**, Jie Wang**[1,2] *****, Qijie Peng**[1,2]**, Ruibo Guo**[2]**, Guoping Wu**[2]**, Bin Li**[2]
[1]MoE Key Laboratory of Brain-inspired Intelligent Perception and Cognition
[2]University of Science and Technology of China
`{yr0013, sa24006073, ruiboguo, guoping}@mail.ustc.edu.cn`
`{jiewangx, binli}@ustc.edu.cn`

## Abstract

Generalization in visual reinforcement learning (VRL) aims to learn agents that can adapt to test environments with unseen visual distractions. Despite advances in robust representations learning, many methods do not take into account the essential downstream task of sequential decision-making. This leads to representations that lack critical long-term information, impairing decision-making abilities in test environments. To tackle this problem, we propose a novel *robust action-value representation learning* (ROUSER) under the information bottleneck (IB) framework. ROUSER learns robust representations to capture long-term information from the decision-making objective (i.e., action values). Specifically, ROUSER uses IB to encode robust representations by maximizing their mutual information with action values for long-term information, while minimizing mutual information with state-action pairs to discard irrelevant features. As action values are unknown, ROUSER proposes to decompose robust representations of state-action pairs into one-step rewards and robust representations of subsequent pairs. Thus, it can use known rewards to compute the loss for robust representation learning. Moreover, we show that ROUSER accurately estimates action values using learned robust representations, making it applicable to various VRL algorithms. Experiments demonstrate that ROUSER outperforms several state-of-the-art methods in eleven out of twelve tasks, across both unseen background and color distractions.

## 1 Introduction

Generalization in visual reinforcement learning (VRL) has received considerable attention (Mnih et al., 2015; Yarats et al., 2021; Kirk et al., 2021; Zhu et al., 2023) due to its potential to learn agents that can address complex tasks across diverse environments in real-world applications, e.g., autonomous driving (Zhou et al., 2023b; Cao et al., 2023), robot control (Xing et al., 2021; Liu et al., 2023b), and chip design (Geng et al., 2024; Wang et al., 2024c; 2025b). It refers to agents' capability to directly use their learned skills to unknown environments, where visual distractions (e.g., dynamic backgrounds or object colors under control) may differ from those encountered during training (Li et al., 2021; Ali et al., 2023). Thus, the generalizable agents can execute tasks with high performance when encountering environments under unseen distractions without extensive retraining.

To learn generalizable agents, one of promising approaches in VRL aims to develop robust representations against visual distractions in environments (Kemertas & Aumentado-Armstrong, 2021; Mazoure et al., 2022). Specifically, some methods introduce data augmentations (Hansen & Wang, 2021; Huang et al., 2022) and contrastive learning (Laskin et al., 2020; Kim et al., 2021) to learn representations that are robust to irrelevant and spurious information. Other methods apply metric learning (Zhang et al., 2021; Lopez et al., 2022) to encode task-relevant information for robust representation learning. Then, based on the learned robust representations, these methods can directly use traditional VRL algorithms (Schulman et al., 2017; Fujimoto et al., 2018; Haarnoja et al., 2018)

---

*Corresponding author. Email: jiewangx@ustc.edu.cn.

to optimize the objectives of the downstream task of sequential decision-making, i.e., maximizing the expected cumulative rewards (i.e., action values) over these environments.

However, the aforementioned representation learning approaches often do not take into account the essential downstream decision-making. This results in the representations that cannot capture critical long-term robust information in sequential data, which is a key factor for generalization in VRL (Zhou et al., 2022; Qi et al., 2022; Wang et al., 2023a). Although such representations are robust against visual distractions, they cannot facilitate the generalization of sequential decision-making.

In this paper, we propose a *robust action-value representation learning* (ROUSER) for VRL generalization under the information bottleneck (IB). ROUSER introduces IB to learn robust representations that capture long-term information from the sequential decision-making objective (i.e., action values).

Specifically, ROUSER first applies IB to propose a *robust action-value representation*, which is correlated with the action value. This robust representation for each state-action pair maximizes mutual information with the action value, while minimizing mutual information with the associated state-action pair. Thus, this representation can capture long-term information without irrelevant features. Then, as the true action values are unknown, we cannot directly predict action values to learn the robust action-value representation. Inspired by task decomposition methods (Dietterich, 2000; Russell & Zimdars, 2003; Dayan, 1993; Barreto et al., 2017), ROUSER proposes to decompose the robust action-value representation into a *compressed reward representation*—which captures only information from a one-step reward—and a subsequent robust action-value representation. With this recursive form, ROUSER can use the known one-step rewards instead of unknown action values to compute the loss associated with robust action-value representations.

Moreover, for compressed reward representations, ROUSER introduces a reward model in the IB framework. This model maximizes mutual information between one-step rewards and compressed reward representations, while minimizing it between such representations and associated state-action pairs. Therefore, ROUSER can use one-step rewards and state-action pairs to learn compressed reward representations that encode only information from rewards.

This study proposes robust action-value representations in the IB framework, capturing long-term information during the robust representation learning process. Thus, it significantly enhances performance in sequential decision-making against unseen visual distractions. In addition, we provide a theoretical guarantee for ROUSER, establishing a bound between the true action-value function and the action-value function on top of learned robust action-value representations. This shows that ROUSER can accurately estimate true action values using learned robust action-value representations. Building upon this proof, we present an advantage of ROUSER, which is its applicability to various VRL algorithms to improve generalization. We can integrate ROUSER with the VRL critic for action-value estimation, using a critic's embedding as a robust action-value representation. Experiments in Section 5.2 demonstrate the applicability by combining ROUSER with traditional VRL algorithms, including policy gradient (Yarats et al., 2022) and value-based (Hosu & Rebedea, 2016) methods.

It is worth noting that our study differs from task decomposition methods, which learn the representations from action values but do not take into account robustness against visual distractions. We use one-step rewards to extract long-term information from action values and leverage IB to discard irrelevant features in VRL. We summarize the major contributions below.

- To the best of our knowledge, this study is the first to learn representations from action values against visual distractions. It encodes robust and long-term information rather than just robust features to facilitate downstream decision-making for generalization.
- We show that ROUSER can accurately estimate action values using the learned robust action-value representations. With this proof, we can integrate ROUSER with various VRL algorithms to estimate the action values, thereby enhancing robustness.
- Extensive experiments demonstrate that ROUSER outperforms several state-of-the-art VRL methods in eleven out of twelve tasks across both unseen background and color distractions.

## 2  RELATED WORK

**Representation Learning for Generalization in VRL.**  Representation learning approaches for generalization in VRL aim to improve agents' performance on unseen test environments by extracting

robust representations from training environments. Some approaches design an auxiliary contrastive task to maximize the mutual information between representations from similar observations while minimizing the mutual information between representations from dissimilar observations (Kim et al., 2019; Laskin et al., 2020; Fan & Li, 2022). To obtain similar observations, they always leverage data augmentations to generate observations (Hansen et al., 2021; Hansen & Wang, 2021; Huang et al., 2022; Ye et al., 2023; Wang et al., 2025a). However, they often do not take into account the downstream task of sequential decision-making, leading to representations that lack long-term information. Some approaches introduce the bisimulation metric with a recursive form (Ferns & Precup, 2014; Castro, 2020; Agarwal et al., 2021) to identify robust representations using long-term information, similar to our approach. However, it is difficult to directly compute the recursive metric. Thus, for implementation, they often simplify the computation to capture information from the immediate future. Other methods directly predict reward sequences to learn robust representations (Yang et al., 2022; Zhou et al., 2023a), which captures long-term task-relevant information for generalization. Instead of using reward sequences, our approach simply applies one-step rewards to encode the sequential information, and we further introduce the IB principle to improve robustness.

**Task Decomposition in RL.** We use the *task decomposition* to represent methods that decompose a task into independent sub-tasks, so that each task has the same environment dynamics but a different reward function. Then, they propose to learn action-value representations invariant to different tasks/reward functions. Some of them, namely *successor feature* methods (Dayan, 1993; Barreto et al., 2017; 2018; 2020), use such representations to generalize to different tasks for transfer learning. Others, namely *value decomposition* methods (Sutton et al., 1999; Russell & Zimdars, 2003; Makino, 2023), apply such representations to improve sample efficiency in new tasks. See Appendix A for details. Motivated by their framework where action-value representations can be decomposed into an infinite sequence of reward representations, we (1) learn the action-value representations for long-term information using one-step rewards and (2) filter out irrelevant information from the one-step reward representations to improve robustness of overall action-value representations.

It is worth noting that in this article, we aim to learn agents in training environments to generalize to test environments with unseen visual distractions, while remaining a consistent reward function. This differs from task decomposition methods, which primarily extract knowledge invariant to tasks between tasks with varying reward functions and no visual distractions. Moreover, task decomposition methods can easily encode irrelevant features in the setting of this article, as they learn representations from environment dynamics, which involve visual distractions in VRL (Yang et al., 2022).

**Information Bottleneck (IB).** The IB principle in supervised learning (Tishby & Zaslavsky, 2015; Saxe et al., 2018) aims to learn compressed representations including minimal information relevant to downstream tasks, improving generalization performance. Specifically, it regularizes representations by minimizing the mutual information between inputs and representations, while maximizing the mutual information between representations and labels. In VRL, recent methods (Pei & Hou, 2019; Xiang et al., 2023) also focus on the IB principle that can learn compressed representations to improve generalization capabilities of agents. Different from supervised learning, in VRL, these methods only have one-step rewards but action values, i.e., the targets in VRL corresponding to labels in supervised learning. Thus, some of them (Federici et al., 2020; Fan & Li, 2022) follow the unsupervised learning paradigm, using the IB framework to regularize representations from similar inputs or multi-view image observations. Since the unsupervised learning in the IB framework is more challenging than supervised learning (Federici et al., 2020), we use one-step rewards—the known supervised signals in RL—to extract their compressed representations for robust action-value representation learning.

## 3 PRELIMINARIES

**Visual RL.** We consider a family of environments $\mathcal{E}$. Each environment $e \in \mathcal{E}$ is a block Markov decision process (BMDP) (Zhang et al., 2020) denoted by $\mathcal{M}^e = \langle \mathcal{S}, \mathcal{O}^e, \mathcal{A}, \mathcal{R}, p, p^e, \gamma \rangle$. Here, $\mathcal{S}$ is the state space, $\mathcal{O}^e$ is the observation space in $e$, $\mathcal{A}$ is the action space, $\mathcal{R}$ is the reward space, $p(s', r \mid s, a)$ is the state transition probability, $p^e(o', r \mid o, a)$ is the observation transition probability (i.e., environment dynamics) varying with the environment $e \in \mathcal{E}$, and $\gamma \in [0, 1)$ is the discount factor. For simplicity, we use bold letters (e.g., $\mathbf{o}$ and $\mathbf{a}$) to denote random variables, normal letters (e.g., $o$ and $a$) to denote samples, and $\mathcal{O}$ to denote the set of $\mathcal{O}^e$ for all $e \in \mathcal{E}$.

We suppose that an agent reaches an unseen latent state $s$ and obtains an observation $o$ on an environment $e \in \mathcal{E}$. In a BMDP, the observation is determined by a state and some task-irrelevant visual factors varying with environments, e.g., backgrounds or agent colors in decision-making process. Each state $s$ does not involve irrelevant features in VRL and is invariant to the family of environments. Formally, let $\mathcal{X}$ be the set of such visual factors. We introduce an observation function $g : \mathcal{S} \times \mathcal{X} \to \mathcal{O}$ (Zhang et al., 2020; Song et al., 2020) such that $\mathbf{o} = g(\mathbf{s}, \mathbf{x})$. Here, $\mathbf{x}$ is a random variable in $\mathcal{X}$, which is independent of $\mathbf{s}$ and $\mathbf{a}$ with a specific transition probability $q^e(x' \mid x)$.

Moreover, we assume that the environments follow a generalized Block structure Zhang et al. (2020). That is, an observation $o \in \mathcal{O}$ uniquely determines its generating state $s$ and the visual factor $x$. This assumption implies that the observation function $g(s, x)$ is invertible with respect to both $s$ and $x$. Then, we have $p^e(o', r \mid o, a) = p(s', r \mid s, a) q^e(x' \mid x)$. Note that the expected reward $r(o, a) = r(s, a) = \mathbb{E}_p[\mathbf{r} \mid s, a]$ depends on the corresponding *state-action pair* rather than the observation-action pair, where $\mathbf{r}$ is the random variable of the one-step reward.

We aim to learn an agent with a policy $\pi : \mathcal{O} \to \Delta(\mathcal{A})$ that maximizes the expected cumulative reward $\mathbb{E}^e\left[\sum_{t=0}^{\infty} \gamma^t r_t\right]$ simultaneously in all $e \in \mathcal{E}$, where $\mathbb{E}^e[\cdot]$ means that the expectation is taken in $e$. We use $Q^\pi(o_t, a_t) = Q^\pi(s_t, a_t) = \mathbb{E}_{\pi, p}\left[\sum_{i=0}^{\infty} \gamma^i r(o_{t+i}, a_{t+i})\right]$ to denote the action values.

## 4 ALGORITHM

In this paper, we propose to learn robust action-value representations that capture long-term information for decision-making. Then, we introduce robust and compressed reward representations and use temporal-difference (TD) learning (Sutton & Barto, 2018) paradigm to learn robust action-value representations, guided by an IB-based objective to ensure robustness throughout the learning process.

### 4.1 ROBUST ACTION-VALUE REPRESENTATIONS IN THE IB FRAMEWORK

In this subsection, we propose robust action-value representations in the IB framework.

Firstly, motivated by task decomposition methods, we provide the *action-value representation* in Definition 4.1. This action-value representation is linearly correlated with the action-value function following a given policy, containing the long-term information from the action-value function. Therefore, we propose using the action-value representation to facilitate the agent's decision-making performance in VRL test environments.

**Definition 4.1.** A representation in a $D$-dimensional space $\mathcal{Z}$ is an *action-value representation* $H^\pi(o_t, a_t)$ if there exists a linear mapping $\Phi : \mathcal{Z} \to \mathbb{R}$ such that

$$Q^\pi(o_t, a_t) = \Phi\big(H^\pi(o_t, a_t)\big) = \big\langle \omega, H^\pi(o_t, a_t) \big\rangle, \quad \forall o_t \in \mathcal{O}, a_t \in \mathcal{A}, \tag{1}$$

where the dimension $D > 1$, $\langle \cdot, \cdot \rangle$ denotes the inner product, and $w$ denotes the weight vector of $\Phi$.

Then, for VRL generalization, due to visual factors $x_t \in \mathcal{X}$, the action-value representation may involve irrelevant information. This motivates us to introduce the IB principle, compressing the action-value representation to discard irrelevant information. Specifically, we employ the conditional IB (Chechik & Tishby, 2002) to provide Definition 4.2. In this definition, we propose a *robust action-value representation*, the action-value representation for each observation-action pair maintains maximum mutual information with the corresponding action value while minimizing mutual information with the observation-action pair given the action value.

**Definition 4.2.** An action-value representation is a *robust action-value representation* $Z^\pi(o_t, a_t) = \mathbb{E}[\mathbf{Z}_t]$ such that for all $o_t \in \mathcal{O}$ and $a_t \in \mathcal{A}$,

$$\mathbf{Z}_t = \arg\min_{\mathbf{H}_t} \mathcal{I}(\mathbf{u}_t; \mathbf{H}_t \mid \mathbf{Q}_t) - \beta \mathcal{I}(\mathbf{Q}_t; \mathbf{H}_t), \tag{2}$$

where $\mathbf{H}_t$ is the random variable of $H^\pi(o_t, a_t)$, $\mathbf{Q}_t$ is the random variable of $Q^\pi(o_t, a_t)$, $\mathbf{u}_t$ is the random variable of $(o_t, a_t)$, $\mathcal{I}(\mathbf{y}_1; \mathbf{y}_2)$ is the mutual information between random variables $\mathbf{y}_1$ and $\mathbf{y}_2$, $\mathcal{I}(\mathbf{y}_1; \mathbf{y}_2 \mid \mathbf{y}_3)$ is the conditional mutual information to quantify the information between $\mathbf{y}_1$ and $\mathbf{y}_2$ given $\mathbf{y}_3$, and $\beta$ is a hyperparameter.

However, it is worth noting that we do not have the true action values during the training process of VRL, and we thus cannot directly use Equations 1 and 2 to learn robust action-value representations.

### 4.2 COMPRESSED REWARD REPRESENTATIONS FROM ONE-STEP REWARDS

To address the aforementioned challenge posed by unknown action values, we propose using one-step rewards instead. We first present how to learn simple action-value representations, and we then discuss how to learn our proposed robust action-value representations in this section.

**The Recursive Form of Action-Value Representations and the Reward Representations.** To learn simple action-value representations, following task decomposition methods, we introduce a reward representation $h \in \mathcal{Z}$ that is linearly related to the one-step reward, with a linear mapping $f : \mathcal{Z} \to \mathbb{R}$ and the weight vector $\omega'$.

$$r(o_t, a_t) = f\big(h(o_t, a_t)\big) = \langle \omega', h(o_t, a_t) \rangle. \tag{3}$$

Based on the reward representations, we can derive the action-value representations such that

$$H^\pi(o_t, a_t) = \mathbb{E}_{\pi,p}\left[ \sum_{i=0}^{\infty} \gamma^i h(o_{t+i}, a_{t+i}) \right], \quad Q^\pi(o_t, a_t) = \langle \omega', H^\pi(o_t, a_t) \rangle. \tag{4}$$

Equation 4 shows that the action-value representation is linearly correlated with the infinite sequence of reward representations. We provide the details in Appendix C.1.

With Equation 4, we can derive a recursive form, i.e., the action-value representation for each observation-action pair consists of a reward representation and the action-value representation of its subsequent observation-action pair.

$$H^\pi(o_t, a_t) = h(o_t, a_t) + \gamma \mathbb{E}_{\pi,p}\left[ H^\pi(o_{t+1}, a_{t+1}) \right]. \tag{5}$$

With the linear relation in Equation 3 and the recursive form in Equation 5, we can simply compute the loss for learning such action-value representations. Moreover, we show that using Equation 5 can converge in tabular settings in Appendix C.2.

**Compressed Reward Representations for Robust Action-Value Representations.** Based on Definition 4.2, Equations 4, and 5, as the action-value representations are decomposed into an infinite sequence of reward representations, which serve as their foundation, we can directly filter out task-irrelevant information from such reward representations to improve robustness of *overall* action-value representations, learning robust action-value representations without irrelevant features. Thus, we propose using the IB framework to regulate the reward representations.

Since (1) the one-step reward is task-relevant without irrelevant features, and (2) the observation may involve irrelevant visual factors, we propose a *compressed reward representation* in Definition 4.3. This representation for each observation-action pair is linearly related to the corresponding one-step reward and does not involve any irrelevant information by preserving maximum mutual information with this one-step reward while minimizing mutual information with its observation-action pair.

**Definition 4.3.** A reward representation is a *compressed reward representation* $z(o_t, a_t) = \mathbb{E}\left[ \mathbf{z}_t \right]$ such that for all $o_t \in \mathcal{O}$ and $a_t \in \mathcal{A}$,

$$\mathbf{z}_t = \underset{\mathbf{h}_t}{\arg\min}\, \mathcal{I}\left( \mathbf{u}_t; \mathbf{h}_t | \mathbf{r}_t \right) - \beta \mathcal{I}(\mathbf{r}_t; \mathbf{h}_t), \tag{6}$$

where $\mathbf{h}_t$ is the random variable of $h(o_t, a_t)$, and $\beta$ is a hyperparameter.

In Equation 6, given the one-step reward $\mathbf{r}_t$, the conditional mutual information $\mathcal{I}(\mathbf{u}_t; \mathbf{h}_t | \mathbf{r}_t)$ quantifies information in $\mathbf{h}_t$ that is irrelevant. The mutual information $\mathcal{I}(\mathbf{r}_t; \mathbf{h}_t)$ quantifies task-relevant information shared between the reward $\mathbf{r}_t$ and the representations $\mathbf{h}_t$. $\beta$ is a hyperparameter that determines the preference over the trade-off between task-relevant and task-irrelevant information.

Based on Definition 4.3 and Equation 4, the robust action-value representation is linearly correlated with the infinite sequence of compressed reward representations. Thus, we can also use the recursive form of robust action-value representations, as shown in Equation 5, to learn these representations.

Furthermore, in the following of this part, we discuss *how to compute mutual information on Equation 6*. We provide detail formulas of Equation 6 given the joint distribution and conditional joint distribution of $\mathbf{u}_t$, $\mathbf{z}_t$, and $\mathbf{r}_t$. Note that for simplicity, we use $\mathbf{z}$ instead of $\mathbf{h}$.

$$\mathcal{I}(\mathbf{u}_t; \mathbf{z}_t | \mathbf{r}_t) = \mathbb{E}_{\mathbf{u}_t, \mathbf{z}_t, \mathbf{r}_t}\left[ \log \frac{p(\mathbf{z}_t | \mathbf{u}_t)}{p(\mathbf{z}_t | \mathbf{r}_t)} \right], \quad \mathcal{I}(\mathbf{r}_t; \mathbf{z}_t) = \mathbb{E}_{\mathbf{r}_t, \mathbf{z}_t}\left[ \log \frac{p(\mathbf{r}_t | \mathbf{z}_t)}{p(\mathbf{r}_t)} \right]. \tag{7}$$

To estimate the probability density $p(\mathbf{z}_t|\mathbf{u}_t)$ on Equation 7, we assume that a compressed reward representation $z(\mathbf{o}_t, \mathbf{a}_t)$ for any observation-action pair $(\mathbf{o}_t, \mathbf{a}_t)$ follows a Gaussian distribution $\mathcal{N}_\psi$, and we provide the detailed implementation in the next subsection.

However, due to the modelling of $\mathbf{z}_t$, the approximation of $p(\mathbf{z}_t|\mathbf{r}_t)$ is intractable. Inspired by the method that estimates conditional mutual information (Alemi et al., 2017; Fan & Li, 2022), we introduce a new representation $\mathbf{k}_t \in \mathcal{Z}$ extracted directly from the one-step reward $\mathbf{r}_t$ to estimate $p(\mathbf{k}_t|\mathbf{r}_t)$ instead of $p(\mathbf{z}_t|\mathbf{r}_t)$. Thus, we can derive an upper bound of the conditional mutual information.

$$\mathcal{I}(\mathbf{u}_t; \mathbf{z}_t|\mathbf{r}_t) \leq \mathcal{D}_{\mathcal{KL}}\Big(p(\mathbf{z}_t|\mathbf{u}_t)\|p(\mathbf{k}_t|\mathbf{r}_t)\Big). \tag{8}$$

We provide details in Appendix C.3. We then can minimize the upper bound—i.e., the Kullback-Leibler (KL) divergence between $p(\mathbf{z}_t|\mathbf{u}_t)$ and $p(\mathbf{k}_t|\mathbf{r}_t)$—to discard irrelevant and redundant information for compressed reward representation learning. Moreover, the representation $\mathbf{k}_t$ is directly mapped from the one-step reward $\mathbf{r}_t$. To prevent this mapping from introducing potential redundant information into $\mathbf{k}_t$, we first use $\mathcal{I}(\mathbf{r}_t; \mathbf{k}_t|\mathbf{u}_t)$ to quantify this information and then minimize $\mathcal{I}(\mathbf{r}_t; \mathbf{k}_t|\mathbf{u}_t)$. Therefore, we have the whole optimization objective to minimize for compressed reward representation learning.

$$\mathcal{L}_{\mathrm{cib}} = \mathcal{I}(\mathbf{u}_t; \mathbf{z}_t|\mathbf{r}_t) - \beta\mathcal{I}(\mathbf{r}_t; \mathbf{z}_t) + \mathcal{I}(\mathbf{r}_t; \mathbf{k}_t|\mathbf{u}_t). \tag{9}$$

## 4.3 ROUSER ALGORITHM

Here, we present the detailed algorithm of ROUSER. Firstly, we provide the losses for compressed reward representations. Then, we introduce the loss with TD learning paradigm for robust action-value representation learning. Moreover, we provide the theoretical analysis of learned robust action-value representations. We provide the architecture and pseudocode of ROUSER in Appendix D.

**IB-based Losses for Compressed Reward Representation Learning.** The optimization objective for learning compressed reward representations consists of two parts: a *prediction loss* based on Equation 3 and a *conditional IB loss* based on Equation 9.

Firstly, we introduce a *reward model* $\psi$ to encode the compressed reward representation from an observation-action pair and assume that a compressed reward representation $z(o_t, a_t) = \mathbb{E}[\mathbf{z}_t]$ for any observation-action pair $(o_t, a_t)$ follows a Gaussian distribution $\mathcal{N}_\psi = \mathcal{N}\big(z_\psi(o_t, a_t), \Sigma_\psi(o_t, a_t)\big)$, where $z_\psi(o_t, a_t)$ is the mean and $\Sigma_\psi(o_t, a_t)$ is the diagonal covariance. We also provide a parameterized model $\eta$ to estimate the representation $k(r_t) = \mathbb{E}[\mathbf{k}_t]$. $\mathbf{k}_t$ is drawn from the Gaussian distribution $\mathcal{N}_\eta = \mathcal{N}\big(k_\eta(r_t), \Sigma_\eta(r_t)\big)$, where $k_\eta(r_t)$ is the mean of $\mathbf{k}_t$, and $\Sigma_\eta(r)$ is the diagonal covariance.

Secondly, we provide the prediction loss based on Equation 3 to maintain the linear mapping relationship between the compressed reward representations and corresponding one-step rewards. Building upon model-based RL methods (Janner et al., 2019; Wang et al., 2022; 2023c) that model one-step rewards as a Gaussian distribution, we predict the mean and the standard deviation of reward distribution to learn reward representations. We formulate the prediction loss based on KL divergence:

$$\mathcal{L}_{\mathrm{pred}}(\psi, \omega', \omega'') = \mathbb{E}_{(\mathbf{o}_t, \mathbf{a}_t, \mathbf{r}_t) \sim \mathcal{B}}\left[\frac{\left\|\langle\omega', z_\psi(\mathbf{o}_t, \mathbf{a}_t)\rangle - \mathbf{r}_t\right\|_2^2}{2 \cdot \langle\omega'', z_\psi(\mathbf{o}_t, \mathbf{a}_t)\rangle^2} + \log\langle\omega'', z_\psi(\mathbf{o}_t, \mathbf{a}_t)\rangle\right], \tag{10}$$

where $\omega'$ denote the weight vector mentioned in Equation 3 for the computation of the mean of one-step rewards, $\omega''$ denote the weight vector to compute the standard deviation of one-step rewards, and $\mathcal{B}$ denotes the replay buffer.

Finally, we propose the conditional IB loss based on Equation 9 to learn compressed reward representations, discarding task-irrelevant information while preserving task-relevant information. With the models $\psi$ and $\eta$, we can reformulate Equation 9 as:

$$\mathcal{L}_{\mathrm{info}}(\psi, \eta) = \mathbb{E}_{\mathcal{B}}\left[-\mathcal{I}(\mathbf{z}_t; \mathbf{k}_t) + \alpha\mathcal{D}_{\mathcal{SKL}}\Big(\mathcal{N}_\psi\|\mathcal{N}_\eta\Big)\right], \tag{11}$$

where $\alpha$ derived from $\beta$ is a hyperparameter that is tuned by an exponential scheduler during training, and $\mathcal{D}_{\mathcal{SKL}}$ represents the symmetrized KL divergence (Jeffreys, 1998) obtained by averaging the KL divergences $\mathcal{D}_{\mathcal{KL}}(\mathcal{N}_\psi\|\mathcal{N}_\eta)$ and $\mathcal{D}_{\mathcal{KL}}(\mathcal{N}_\eta\|\mathcal{N}_\psi)$. Please refer to Appendix C.3.

The first term of $\mathcal{L}_{\text{info}}$ in Equation 11 maximizes mutual information between the representation $\mathbf{z}_t$ and the one-step reward $\mathbf{r}_t$. This term in fact maximizes task-relevant information in the representation $\mathbf{z}_t$. The second term of $\mathcal{L}_{\text{info}}$ minimizes the conditional mutual information between the observation-action pair $(\mathbf{o}_t, \mathbf{a}_t)$ and the representation $\mathbf{z}_t$. It represents that we can minimize irrelevant and redundant information in the representation $\mathbf{z}_t$. See Appendix C.3 for the derivation of Equation 11.

Further, to estimate the mutual information $\mathcal{I}(\mathbf{z}_t; \mathbf{k}_t)$ between $\mathbf{z}_t$ and $\mathbf{k}_t$ for a transition $(\mathbf{o}, \mathbf{a}, \mathbf{r}) \sim \mathcal{B}$, we can leverage the InfoNCE (van den Oord et al., 2018; Laskin et al., 2020) function. Thus, we have

$$\mathcal{I}(\mathbf{z}_t; \mathbf{k}_t) = -\log \frac{\exp(\text{sim}(\mathbf{z}_t, \mathbf{k}_t^+))}{\exp(\text{sim}(\mathbf{z}_t, \mathbf{k}_t^+)) + \sum_{i=1}^{N-1} \exp(\text{sim}(\mathbf{z}_t, \mathbf{k}_{t,i}))}, \tag{12}$$

where $\text{sim}(\mathbf{x}, \mathbf{y}) = \mathbf{x}^T \mathbf{y}$, $N$ is the batch size.

**TD-Based Loss for Robust Action-Value Representation Learning.**   As the robust action-value representation has a recursive nature, we employ the TD learning paradigm—the commonly used technique to learn an objective with a recursive form—to learn the representation.

Based on Equations 2, 5, and 6, we derive a recursive form of the robust action-value representation.

$$Z^\pi(\mathbf{o}_t, \mathbf{a}_t) = z(\mathbf{o}_t, \mathbf{a}_t) + \gamma \mathbb{E}_{\pi,p} \left[ Z^\pi(\mathbf{o}_{t+1}, \mathbf{a}_{t+1}) \right], \tag{13}$$

where $Z^\pi$ and $z$ subject to Equations 2 and 6, respectively.

Then, we model the critic using the parameter $\theta$ to estimate the action values. For robust action-value representation learning, we *do not* change critic's structure and *directly* use embeddings from critic's center layer as robust action-value representations (see Figure 1). Therefore, we also parameterize the robust action-value representations as $Z_\theta$. Based on Equation 13, we have the TD-based loss as:

$$\mathcal{L}_{\text{robust}}(\theta) = \mathbb{E}_\mathcal{B} \left[ \left\| Z_\theta(\mathbf{o}_t, \mathbf{a}_t) - \hat{Z}(\mathbf{o}_t, \mathbf{a}_t) \right\|_2^2 \right], \ \hat{Z}(\mathbf{o}_t, \mathbf{a}_t) = z_\psi(\mathbf{o}_t, \mathbf{a}_t) + \gamma \mathbb{E}_\pi \left[ Z_{\bar{\theta}}(\mathbf{o}_{t+1}, \mathbf{a}_{t+1}) \right], \tag{14}$$

where $(\mathbf{o}_t, \mathbf{a}_t, \mathbf{r}_t, \mathbf{o}_{t+1}) \sim \mathcal{B}$, and $\bar{\theta}$ denotes the parameter of the target critic that are obtained as an exponentially moving average of the parameter $\theta$ of the critic (Mnih et al., 2015). Moreover, if we apply the soft actor-critic framework (Haarnoja et al., 2018) that adds entropy of the policy to one-step rewards, we will also add this entropy into each dimension of compressed reward representations.

According to Equation 14, we provide a theoretical guarantee for the robust action-value representation learning in the following theorem. It gives a bound between the true action-value function and the action-value function on top of learned robust action-value representations (see Appendix C.4).

**Theorem 4.4.** *Let $Z$ be a learned robust action-value representation from any observation-action pair $(o, a) \in \mathcal{O} \times \mathcal{A}$, $Q_e^\pi : \mathcal{O} \times \mathcal{A} \to \mathbb{R}$ be the true action-value function of a policy $\pi$ in the environment $e \in \mathcal{E}$, $f_e^* : \mathcal{Z} \to \mathbb{R}$ be the optimal linear mapping on the representation space, and $\epsilon$ be a bound of estimation error for each compressed reward representation $z$, i.e., $|f_e^*(z) - r| \le \epsilon$. For any $(o, a) \in \mathcal{O} \times \mathcal{A}$ and $e \in \mathcal{E}$, we have*

$$0 \le \left| Q_e^\pi(o, a) - f_e^*\big(Z(o, a)\big) \right| \le \frac{1}{1 - \gamma} \epsilon.$$

This theoretical proof allows us to directly use embeddings of the critic's center layer as robust action-value representations for estimating action values. This shows that ROUSER can be integrated with various VRL methods. We combine ROUSER with traditional VRL methods in Section 5.2.

In addition, to alleviate the overestimation bias of action values, Fujimoto et al. (2018) introduces double critics with $\theta_1$ and $\theta_2$. We note that our representation learning based on Equation 14 also suffers from the overestimation bias as well as the action-value function approximation (Fujimoto et al., 2018). Thus, we propose to leverage the overestimation bias of robust action-value representations to alleviate the bias of estimated action values. We reformulate Equation 14 as:

$$\mathcal{L}_{\text{robust}}(\theta_1, \theta_2) = \mathbb{E}_{(\mathbf{o}_t, \mathbf{a}_t, \mathbf{r}_t, \mathbf{o}_{t+1}) \sim \mathcal{B}} \left[ \sum_{i=1,2} \left\| Z_{\theta_i}(\mathbf{o}_t, \mathbf{a}_t) - \hat{Z}(\mathbf{o}_t, \mathbf{a}_t) \right\|_2^2 \right], \tag{15}$$

$$\hat{Z}(\mathbf{o}_t, \mathbf{a}_t) = z_\psi(\mathbf{o}_t, \mathbf{a}_t) + \gamma \mathbb{E}_\pi \left[ \min_{i=1,2} \mathbb{I} \, Z_{\bar{\theta}_i}(\mathbf{o}_{t+1}, \mathbf{a}_{t+1}) \right],$$

$$\min_{i=1,2} \mathbb{I} \, Z_{\bar{\theta}_i}(\mathbf{o}_t, \mathbf{a}_t) = \min_{i=1,2} \mathbb{I}_{\omega' \ge 0} \cdot Z_{\bar{\theta}_i}(\mathbf{o}_t, \mathbf{a}_t) + \max_{i=1,2} \mathbb{I}_{\omega' < 0} \cdot Z_{\bar{\theta}_i}(\mathbf{o}_t, \mathbf{a}_t), \tag{16}$$

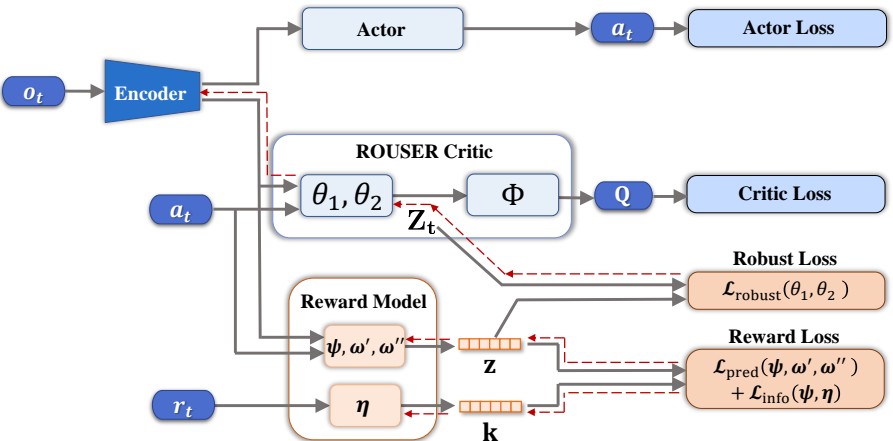

Figure 1: The architecture of ROUSER. We first input $(o_t, a_t, r_t)$ into the reward model to compute the reward loss, updating the reward model. Next, we input $(o_t, a_t)$ into the critic model to compute the robust loss, which updates $(\theta_1, \theta_2)$ to learn robust action-value representations. Moreover, we use traditional actor and critic losses to update the actor and the entire critic model, respectively.

where $\bar{\theta}_1$ and $\bar{\theta}_2$ are parameters of target critics, $\mathbb{I}_{\omega' \geq 0}$ and $\mathbb{I}_{\omega' < 0}$ denote $D$-dimensional vectors of binary values from the indicator function $\mathbb{I}$, and $\omega'$ is the parameter of the linear mapping $f$ equivalent to $\Phi$. With the operator $\min\mathbb{I}$, we select minimums of robust action-value representations positively related to action values and maximums negatively related to action values. Thus, based on the linear relation between action values and robust action-value representations, we can alleviate the overestimation bias in the action-value estimation.

To clearly overview our method, we provide the architecture in Figure 1. It illustrates our proposed two loss functions: the robust loss $\mathcal{L}_{\text{robust}}(\theta_1, \theta_2)$ and the reward loss $\mathcal{L}_{\text{pred}}(\psi, \omega', \omega'') + \mathcal{L}_{\text{info}}(\psi, \eta)$. The detailed implementation of this architecture are presented in Appendix D.

## 5 EXPERIMENTS

We conduct extensive experiments to evaluate the generalization performance of ROUSER on testing environments with unseen visual distractions. Firstly, we demonstrate the effectiveness of ROUSER for solving the visual control tasks on DeepMind Control Suite (DMC, Tassa et al. (2018)) with task-irrelevant visual distractions, including background and color distractions. Then, we assess the applicability of ROUSER by combining it with traditional policy gradient and value-based VRL algorithms. Finally, we visualize the captured long-term information and evaluate the robustness of ROUSER. Moreover, we conduct careful ablation studies to show the effectiveness of ROUSER. We provide additional results of ROUSER in Appendix E.

### 5.1 GENERALIZATION PERFORMANCE ON DMC BENCHMARK WITH CONTINUOUS ACTIONS

We apply visual distractions in DMC benchmark, including color and background distractions (Stone et al., 2021), to investigate the generalization performance of VRL agents. *For color distractions*, we train the agents in a training environment with a weak color change and evaluate in various test environments with strong color changes. It is worth noting that the variance of the color distractions we use is larger than the hard setting proposed in Stone et al. (2021), which poses greater challenges for the agents. *For background distractions*, we follow the hard setting (Agarwal et al., 2021; Yang et al., 2022) to use 2 videos as backgrounds during training and evaluate the generalization performance on 30 unseen videos. See Appendix E.1 for more details. We conduct main experiments on six tasks of DMC. For each task, we train the agents with six random seeds and present means and standard errors of their cumulative rewards at 500,000 environment steps.

We integrate ROUSER with DrQv2 (Yarats et al., 2022) and SRM (Huang et al., 2022) to evaluate the effectiveness of ROUSER on generalization. Note that in **all** experiments under both color and background distractions, ROUSER-DrQv2 consistently outperforms DrQv2, and ROUSER-SRM

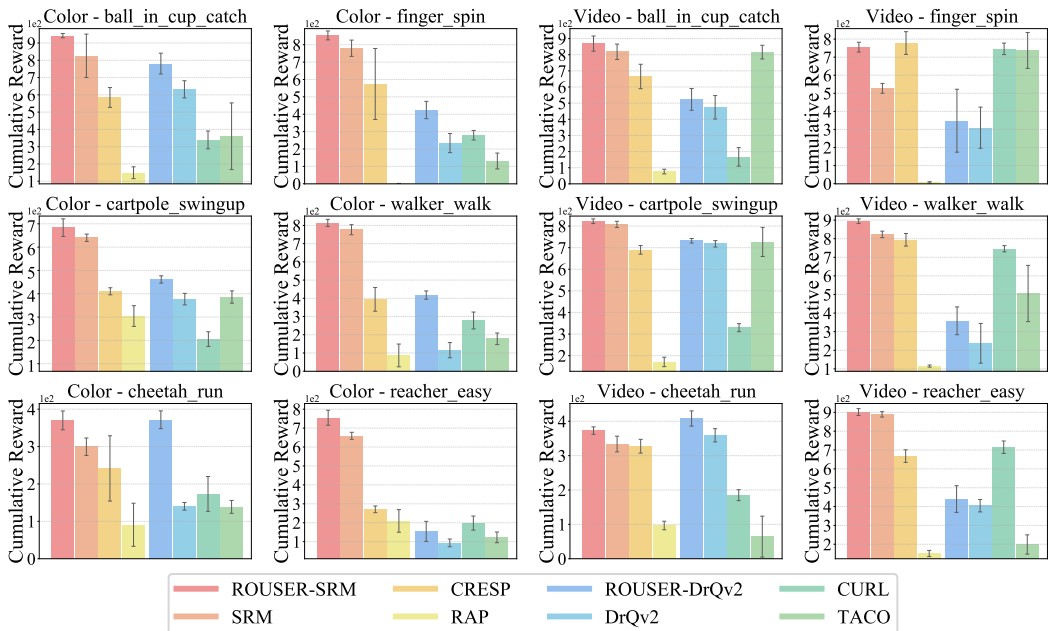

Figure 2: We evaluate ROUSER on DMC with *unseen color* and *video background distractions*. Each result is averaged over 100 episodes at 500K environment steps using six random seeds.

surpasses SRM. We also compare our approach ROUSER against several state-of-the-art (SOTA) methods: (1) CURL (Laskin et al., 2020), (2) RAP (Chen & Pan, 2022), (3) CRESP (Yang et al., 2022), (4) TACO (Zheng et al., 2024). See Appendix E.1 for detailed settings.

Figure 2 illustrates results under color and background distractions, demonstrating that ROUSER outperforms other baselines in **11 out of 12** experiments. See Appendix E.1 for detailed results.

It is worth noting that although ROUSER uses one-step rewards to learn robust action-value representations, it significantly improves generalization performance on tasks with sparse reward functions (e.g., ball_in_cup_catch, finger_spin, and reacher_easy tasks from DMC benchmark). Considering that the actor-critic algorithms (Fujimoto et al., 2018; Haarnoja et al., 2018)—which estimate action values using one-step rewards—also perform well on sparse reward tasks, we believe that our approach ROUSER is also effective in the contexts beyond sparse rewards.

## 5.2 EXTENDING ROUSER TO PROCGEN WITH DISCRETE ACTIONS

In this subsection, motivated by the result in Theorem 4.4, we extend ROUSER to decision tasks with discrete actions, assessing its applicability.

Firstly, Figure 2 shows that ROUSER can enhance generalization when combined with policy gradient VRL algorithms, such as DrQv2 and SRM. Then, to demonstrate that ROUSER can be integrated with value-based methods, we conduct experiments in Procgen (Cobbe et al., 2020), a benchmark with image observations where each game has multiple levels to evaluate generalization. See Appendix E.2 for detailed settings. We integrate ROUSER with QR-DQN (Dabney et al., 2018), a prior SOTA value-based VRL method on Procgen for generalization.

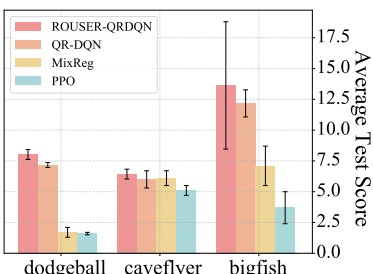

Figure 3: Results on Procgen.

Figure 3 shows Procgen scores on test levels that are averaged over five random seeds with a batch size of 256 at 25M environment steps. The results demonstrate that ROUSER can also improve the generalization performance of agents when combined with the value-based VRL method for discrete control tasks. In addition, we provide the details of these results and the implementation for the combination of ROUSER and QR-DQN in Appendix E.2.

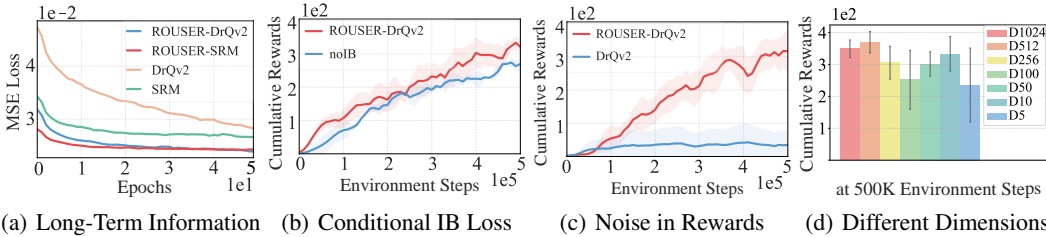

(a) Long-Term Information    (b) Conditional IB Loss    (c) Noise in Rewards    (d) Different Dimensions

Figure 4: We report four results on cheetah_run task with unseen color distractions.

### 5.3 ANALYZING THE REPRESENTATIONS LEARNED BY ROUSER

**Long-Term Robust Information.** We assess how ROUSER can capture long-term robust information. Specifically, we first collect 200K samples under unseen color distractions using a DrQv2's policy. Next, we compute the representations learned by each method from state-action pairs in the collected data. Such representations are extracted from the center layer of the critic in each method. Then, we fix these representations and input them into a 2-layer trainable MLP to predict the future reward sequences over a length of 300. During the prediction process, we use the Adam optimizer with a learning rate of 1e-4. As shown in Figure 4(a), we plot curves of the MSE loss on 1K evaluation samples across three seeds. The lower MSE losses of ROUSER indicate that its learned representations capture long-term information more effectively.

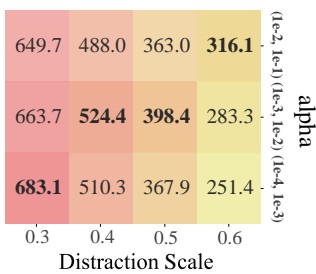

Figure 5: Results with different $\alpha$ in different color scales.

**Effectiveness of IB.** To assess the effectiveness of IB, we remove the conditional IB loss from the objectives of ROUSER. We then train agents to evaluate their generalization performance. Results (i.e., means and standard deviations) over three seeds in Figure 4(b) demonstrate the effectiveness of the conditional IB loss. Moreover, by tuning $\alpha$ in the conditional IB loss from Equation 11, we find that larger values of $\alpha$ result in smaller $\mathcal{D}_{\mathcal{SKL}}$, potentially improving generalization performance. This observation aligns with our analysis in Section 4.3, demonstrating that minimizing $\mathcal{D}_{\mathcal{SKL}}$ (the second term of $\mathcal{L}_{\text{info}}$) effectively discards irrelevant and redundant information. We illustrate the results for different values of $\alpha$ over three seeds in Figure 5. Each result is evaluated at 200K steps on cartpole-swingup task under color distractions (scale from 0.3 to 0.6, see Appendix E.5 for details).

**Noisy One-Step Rewards.** We add noise drawn from $\mathcal{N}(0, 0.1)$ for each reward to show how ROUSER can be robust against noisy rewards. Figure 4(c) demonstrates that ROUSER-DrQv2 significantly outperforms DrQv2, remaining effective under the noisy reward setting.

**Ablation Study for Dimension.** We illustrate results of ROUSER-DrQv2 for different dimensions of robust action-value representations in Figure 4(d). The results, averaged over three seeds on cheetah_run task with color distractions, show that ROUSER is not sensitive to the dimension $D$.

## 6 CONCLUSION

Many existing VRL approaches focus on learning robust representations against irrelevant visual distractions but often do not take into account the long-term decision-relevant information. This may degrade generalization performance of VRL agents. In this paper, we propose robust action-value representations in the IB framework, which encodes long-term information from action values while discarding irrelevant features from image observations during the robust representation learning process. Considering that action values are unknown, we propose to decompose robust action-value representations into infinite sequences associated with rewards. Thus, we can directly use known rewards to learn robust action-value representations. Experiments demonstrate that our approach significantly enhances generalization capabilities in the majority of tasks. We believe this method is general and applicable to other real-word applications, such as end-to-end autonomous driving.

## REPRODUCIBILITY STATEMENT

In this study, to ensure the reproducibility of our approach, we provide key information from the main text and Appendix as follows.

1. **Algorithm.** We provide the architecture and pseudocode of our approach ROUSER in Appendix D. We also provide the detailed implementation of ROUSER in Appendices E.1 and E.2. See Appendix E.5 for the hyperparameters of ROUSER.

2. **Source Code.** According to the architecture and pseudocode in Appendix D, our approach ROUSER can directly build upon traditional VRL algorithms. Specifically, in Section 5.1, we follow Yuan et al. (2023) to use the provided code for DrQv2 and SRM, available at https://github.com/gemcollector/RL-ViGen. In Section 5.2, we apply the code from Wang et al. (2020) for QR-DQN, available at https://github.com/kaixin96/mixreg. Moreover, we implement our codes in Python version 3.7 and make the code available online [1].

3. **Experimental Details.** We provide detailed experiment settings in Section 5.1, Appendices E.1, E.2, and E.3.

4. **Theoretical Proofs.** We provide all proofs in Appendix C.

## ACKNOWLEDGMENTS

The authors would like to thank all the anonymous reviewers for their valuable suggestions. This work was supported by the National Key R&D Program of China under contract 2022ZD0119801 and the National Nature Science Foundations of China grants U23A20388 and 62021001.

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

APPENDIX

## A  RELATED WORK OF TASK DECOMPOSITION

***Value Decomposition.*** Hierarchical RL aims to solve complex tasks into layered sub-goals, enabling agents to tackle challenges at varying levels of abstraction for efficient and scalable learning (Wang et al., 2023b; 2024a; Ling et al., 2024; Kuang et al., 2024; Liu et al., 2024; 2025; 2023a). Some methods suggest that reusing learned skills from simple tasks can effectively improve the sample efficiency in solving the complex composite tasks (Sutton et al., 1999; Dieterich, 2000; Russell & Zimdars, 2003; Makino, 2023). Thus, for a complex sequential decision-making task that can be divided into a series of simple individual sub-tasks, these approaches estimate the action-value function for each sub-task to learn these reusable skills.

Inspired by this hierarchical RL, some methods propose the reward decomposition for composite tasks, which decomposes the one-step rewards into a series of sub-rewards instead of breaking down the composite task (van Seijen et al., 2017; Lin et al., 2019; Grimm & Singh, 2019; Fatemi & Tavakoli, 2022). Specifically, under the assumption that one-step rewards of composite tasks are linear combinations of independent sub-rewards, these methods decompose each one-step reward either manually or automatically (i.e., leveraging deep learning techniques). Then, they combine the action values for each sub-task associated with each sub-reward additively to estimate the overall action-value function, thus efficiently addressing the complex composite task.

In this article, although the reward representations extracted by ROUSER are similar to the sub-rewards from reward decomposition methods, they do not rely on the prior assumption that one-step rewards are linear combinations of independent sub-rewards.

***Successor Features.*** Successor features have garnered attention in the field of transfer RL, which allows flexible knowledge transfer across tasks with different reward functions. Barreto et al. (2017) assume a set of tasks with different reward functions but similar environment dynamics, and thus propose the successor features, which represent the features from environment dynamics involving the long-term sequential information. To learn such features, Barreto et al. (2017; 2018; 2020); Ma et al. (2020); Carvalho et al. (2023) first decouple rewards into the component associated with environment dynamics and the weight vector associated with different tasks. Then, they can use the reward component to learn successor features from the environment dynamics. By using the successor features and the weight vector, they can achieve the transfer learning. Moreover, based on successor features, Abdolshah et al. (2021) propose transferring knowledge across different environments by using a Gaussian distribution to capture the changes in environment dynamics.

Our study differs from these methods, which do not take into account robustness against visual distractions. Also, the successor features learned by the aforementioned methods often involve irrelevant information from environment dynamics in the VRL generalization challenge. Instead, we use one-step rewards to extract long-term information from action values and leverage the IB framework to discard irrelevant features. It is worth NOTING that in all the aforementioned successor feature works, fine-tuning the successor features is essential when encountering unseen testing environments. In contrast, our approach focuses on zero-shot generalization without any fine-tuning.

## B  IMPACT STATEMENTS AND LIMITATIONS

This paper proposes a novel approach called ROUSER to advance the field of generalization in VRL, enhancing the potential of agents in real-world applications. Although our primary focus is on technical innovation, we recognize the potential societal consequences of our work, as generalization in VRL has the potential to influence various domains such as autonomous driving and the financial field. We are committed to ethical research practices and attach great importance to the social implications and ethical considerations in the development of generalization research in VRL.

It is essential for our approach to use the artificially designed rewards for robust action-value representation learning. We look forward to advancing the representation learning in VRL using inverse RL techniques that can derive the reward function from data. This will enable us to introduce a framework that supports autonomic iterative optimization. Moreover, our work may inspire some

studies about the message invariance for message passing-based graph neural networks (Shi et al., 2023; 2025; 2024).

## C  PROOFS

### C.1  PROOF OF THE MAPPING

In this subsection, we prove that the action-value representation is linearly correlated with the infinite sequence of reward representations.

*Proof.* Based on Equations 1, 3, and the definition of the action-value function, we have

$$Q^\pi(o_t, a_t) = \mathbb{E}_{\pi,p} \left[ \sum_{i=0}^\infty \gamma^i r(o_{t+i}, a_{t+i}) \right] \tag{17}$$

$$= \mathbb{E}_{\pi,p} \left[ \sum_{i=0}^\infty \gamma^i \langle \omega', h(o_{t+i}, a_{t+i}) \rangle \right] \tag{18}$$

$$= \left\langle \omega', \mathbb{E}_{\pi,p} \left[ \sum_{i=0}^\infty \gamma^i h(o_{t+i}, a_{t+i}) \right] \right\rangle \tag{19}$$

$$= \langle \omega', H^\pi(o_t, a_t) \rangle. \tag{20}$$

Therefore, we derive an action-value representation $H^\pi(o_t, a_t)$. $\qquad\square$

### C.2  PROOF OF THE CONVERGENCE OF ACTION-VALUE REPRESENTATION LOSS

In this section, we provide a detailed derivation of the convergence of action-value representation. ROUSER leverages the reward representations to introduce an auxiliary objective function with a recursive form. Similar to the TD-learning of action-value functions, we prove that Equations 4 and 5 can be reformulated as a contraction mapping $\mathbf{T}$. Here, we define $Z^\pi$ as a generic function $Z^\pi : \mathcal{O} \times \mathcal{A} \to \mathcal{Z}$, and we define the mapping $\mathbf{T} : \mathcal{Z} \to \mathcal{Z}$.

**Theorem** *There exists a contraction mapping* $\mathbf{T} : \mathcal{Z} \to \mathcal{Z}$ *defined as*
$$(\mathbf{T}Z^\pi)(o_t, a_t) = z(o_t, a_t) + \gamma \cdot \mathbb{E}_{\pi,p} \left[ Z^\pi(o_{t+1}, a_{t+1}) | o_t, a_t \right].$$

*Proof.* Let $\left| Z^\pi(o_t, a_t) \right|_d$ denote the L1 value of the $d$th-dimension of $Z^\pi(o_t, a_t)$, where $1 \leq d \leq D$. Due to the properties of contraction mappings (Sutton & Barto, 2018; Zhou et al., 2023b), we can apply the operator $\mathbf{T}$ to compute the target auxiliary loss function until convergence in tabular settings. $\mathbf{T}$ operator is a contraction in the sup-norm, which we provide as follows.
$$||\mathbf{T}Z_1^\pi - \mathbf{T}Z_2^\pi||_\infty \leq \gamma \cdot ||Z_1^\pi - Z_2^\pi||_\infty. \tag{21}$$
We provide a detailed proof of Equation 21.

$$||\mathbf{T}Z_1^\pi - \mathbf{T}Z_2^\pi||_\infty$$
$$= \max_d \left| z + \gamma \cdot \mathbb{E}_{\pi,p} \left[ Z_1^\pi(o_{t+1}, a_{t+1}) | o_t, a_t \right] - z - \gamma \cdot \mathbb{E}_{\pi,p} \left[ Z_2^\pi(o_{t+1}, a_{t+1}) | o_t, a_t \right] \right|_d$$
$$= \gamma \cdot \max_d \left| \mathbb{E}_{\pi,p} \left[ Z_1^\pi(o_{t+1}, a_{t+1}) | o_t, a_t \right] - \mathbb{E}_{\pi,p} \left[ Z_2^\pi(o_{t+1}, a_{t+1}) | o_t, a_t \right] \right|_d$$
$$= \gamma \cdot \max_d \left| \mathbb{E}_{\pi,p} \left[ Z_1^\pi(o_{t+1}, a_{t+1}) - Z_2^\pi(o_{t+1}, a_{t+1}) | o_t, a_t \right] \right|_d$$
$$\leq \gamma \cdot \max_d \mathbb{E}_{\pi,p} \left[ \left| Z_1^\pi(o_{t+1}, a_{t+1}) - Z_2^\pi(o_{t+1}, a_{t+1}) \right|_d | o_t, a_t \right]$$
$$\leq \gamma \cdot \max_{\substack{d \\ o_{t+1} \in \mathcal{O} \\ a_{t+1} \in \mathcal{A}}} \left| Z_1^\pi(o_{t+1}, a_{t+1}) - Z_2^\pi(o_{t+1}, a_{t+1}) \right|_d$$
$$= \gamma \cdot ||Z_1^\pi - Z_2^\pi||_\infty.$$

$\qquad\square$

Based on Equation 21, we can derive the convergence of auxiliary loss similar to action-value function estimation. So we can apply $\mathbf{T}$ to compute the auxiliary loss until convergence in tabular settings.

### C.3 INFORMATION BOTTLENECK

Here, we provide a derivation of the formulas involved in the IB techniques in our ROUSER approach. We first list the mathematical formulas used in our proof.

**(P.1)** KL Divergence

$$\mathcal{D}_{\mathcal{KL}}\Big(p(\mathbf{x})||q(\mathbf{x})\Big) = \mathbb{E}_{\mathbf{x}}\left[\log\frac{p(\mathbf{x})}{q(\mathbf{x})}\right]. \tag{22}$$

**(P.2)** Mutual Information

$$\mathcal{I}(\mathbf{x};\mathbf{y}) = \mathbb{E}_{\mathbf{x},\mathbf{y}}\left[\log\frac{p(\mathbf{x},\mathbf{y})}{p(\mathbf{x})p(\mathbf{y})}\right] = \mathcal{D}_{\mathcal{KL}}\Big(p(\mathbf{x},\mathbf{y})||p(\mathbf{x})p(\mathbf{y})\Big). \tag{23}$$

**(P.3)** Chain rule of Mutual Information

$$\mathcal{I}(\mathbf{x},\mathbf{y};\mathbf{z}) = \mathcal{I}(\mathbf{y};\mathbf{z}) + \mathcal{I}(\mathbf{x};\mathbf{z}|\mathbf{y}). \tag{24}$$

**(P.4)** Symmetry of Mutual Information

$$\mathcal{I}(\mathbf{x};\mathbf{y}) = \mathcal{I}(\mathbf{y};\mathbf{x}). \tag{25}$$

Then, for Equation 6, we can leverage **(P.2)** to expand the computation of mutual information and conditional mutual information as follows.

$$
\begin{aligned}
\mathcal{I}(\mathbf{r};\mathbf{z}) &= \mathbb{E}_{\mathbf{r},\mathbf{z}}\left[\log\frac{p(\mathbf{r},\mathbf{z})}{p(\mathbf{r})p(\mathbf{z})}\right] \\
&= \mathbb{E}_{\mathbf{r},\mathbf{z}}\left[\log\frac{p(\mathbf{z})p(\mathbf{r}|\mathbf{z})}{p(\mathbf{r})p(\mathbf{z})}\right] \\
&= \mathbb{E}_{\mathbf{r},\mathbf{z}}\left[\log\frac{p(\mathbf{r}|\mathbf{z})}{p(\mathbf{r})}\right].
\end{aligned}
\tag{26}
$$

$$
\begin{aligned}
\mathcal{I}(\mathbf{u};\mathbf{z}|\mathbf{r}) &= \mathbb{E}_{\mathbf{u},\mathbf{z},\mathbf{r}}\left[\log\frac{p(\mathbf{u},\mathbf{z}|\mathbf{r})}{p(\mathbf{u}|\mathbf{r})p(\mathbf{z}|\mathbf{r})}\right] \\
&= \mathbb{E}_{\mathbf{u},\mathbf{z},\mathbf{r}}\left[\log\frac{p(\mathbf{u}|\mathbf{r})p(\mathbf{z}|\mathbf{u},\mathbf{r})}{p(\mathbf{u}|\mathbf{r})p(\mathbf{z}|\mathbf{r})}\right] \\
&= \mathbb{E}_{\mathbf{u},\mathbf{z},\mathbf{r}}\left[\log\frac{p(\mathbf{z}|\mathbf{u},\mathbf{r})}{p(\mathbf{z}|\mathbf{r})}\right] \\
&\overset{*}{=} \mathbb{E}_{\mathbf{u},\mathbf{z},\mathbf{r}}\left[\log\frac{p(\mathbf{z}|\mathbf{u})}{p(\mathbf{z}|\mathbf{r})}\right].
\end{aligned}
\tag{27}
$$

Here in $*$, since $\mathbf{z}$ is the representation of $\mathbf{u}$, we have $p(\mathbf{z}|\mathbf{u},\mathbf{r}) = p(\mathbf{z}|\mathbf{u})$. As discussed in Equation 8, we can introduce a new feature $\mathbf{k}$ to provide an upper bound to estimate the conditional mutual information. The details of the proof in Equation 8 are as follows,

$$
\begin{aligned}
&\mathcal{I}(\mathbf{u};\mathbf{z}|\mathbf{r}) \\
&= \mathbb{E}_{\mathbf{u},\mathbf{z},\mathbf{r}}\left[\log\frac{p(\mathbf{z}|\mathbf{u})}{p(\mathbf{z}|\mathbf{r})}\right] \\
&= \mathbb{E}_{\mathbf{u},\mathbf{z},\mathbf{r}}\left[\log\frac{p(\mathbf{z}|\mathbf{u})}{p(\mathbf{k}|\mathbf{r})}\frac{p(\mathbf{k}|\mathbf{r})}{p(\mathbf{z}|\mathbf{r})}\right] \\
&= \mathbb{E}_{\mathbf{u},\mathbf{z},\mathbf{r}}\left[\log\frac{p(\mathbf{z}|\mathbf{u})}{p(\mathbf{k}|\mathbf{r})}\right] - \mathbb{E}_{\mathbf{u},\mathbf{z},\mathbf{r}}\left[\log\frac{p(\mathbf{z}|\mathbf{r})}{p(\mathbf{k}|\mathbf{r})}\right] \\
&= \mathbb{E}_{\mathbf{r}|\mathbf{u},\mathbf{z}}\left(\mathbb{E}_{\mathbf{u},\mathbf{z}}\left[\log\frac{p(\mathbf{z}|\mathbf{u})}{p(\mathbf{k}|\mathbf{r})}\right]\right) - \mathbb{E}_{\mathbf{u}|\mathbf{z},\mathbf{r}}\left(\mathbb{E}_{\mathbf{z},\mathbf{r}}\left[\log\frac{p(\mathbf{z}|\mathbf{r})}{p(\mathbf{k}|\mathbf{r})}\right]\right) \\
&= \mathbb{E}_{\mathbf{u},\mathbf{z}}\left[\log\frac{p(\mathbf{z}|\mathbf{u})}{p(\mathbf{k}|\mathbf{r})}\right] - \mathbb{E}_{\mathbf{z},\mathbf{r}}\left[\log\frac{p(\mathbf{z}|\mathbf{r})}{p(\mathbf{k}|\mathbf{r})}\right] \\
&= \mathcal{D}_{\mathcal{KL}}\Big(p(\mathbf{z}|\mathbf{u})||p(\mathbf{k}|\mathbf{r})\Big) - \mathcal{D}_{\mathcal{KL}}\Big(p(\mathbf{z}|\mathbf{r})||p(\mathbf{k}|\mathbf{r})\Big) \\
&\leq \mathcal{D}_{\mathcal{KL}}\Big(p(\mathbf{z}|\mathbf{u})||p(\mathbf{k}|\mathbf{r})\Big) = \mathcal{D}_{\mathcal{KL}}\Big(\mathcal{N}_{\psi}||\mathcal{N}_{\eta}\Big).
\end{aligned}
\tag{28}
$$

Since the features $\mathbf{z} \sim \mathcal{N}_\psi$ and $\mathbf{k} \sim \mathcal{N}_\eta$, we can directly use the Gaussian distribution $\mathcal{N}$ to replace the probability density function $p$. Note that the equality holds if the two distributions coincide. Similar to $\mathcal{I}(\mathbf{u}; \mathbf{z}|\mathbf{r})$, $\mathcal{I}(\mathbf{r}; \mathbf{k}|\mathbf{u})$ is upper bounded by $\mathcal{D}_{\mathcal{KL}}\big(\mathcal{N}_\eta||\mathcal{N}_\psi\big)$.

We can also leverage $\mathbf{k}$ to provide a lower bound for the mutual information:

$$\mathcal{I}(\mathbf{r}; \mathbf{z}) \overset{\text{(P.3)}}{=} \mathcal{I}(\mathbf{k}, \mathbf{r}; \mathbf{z}) - \mathcal{I}(\mathbf{z}; \mathbf{k}|\mathbf{r}) \tag{29}$$

$$\overset{*}{=} \mathcal{I}(\mathbf{k}, \mathbf{r}; \mathbf{z}) \tag{30}$$

$$\overset{\text{(P.3)}}{=} \mathcal{I}(\mathbf{k}; \mathbf{z}) + \mathcal{I}(\mathbf{z}; \mathbf{r}|\mathbf{k}) \tag{31}$$

$$\geq \mathcal{I}(\mathbf{k}; \mathbf{z}). \tag{32}$$

Since $\mathbf{k}$ is the representation of $\mathbf{r}$, we have $\mathcal{I}(\mathbf{z}; \mathbf{k}|\mathbf{r}) = 0$, and we thus derive the equality with * of Equation 30. Moreover, the equality in Equation 32 holds if $\mathcal{I}(\mathbf{z}; \mathbf{r}|\mathbf{k}) = 0$, i.e., the representation $\mathbf{k}$ captures all the information associated with the reward $\mathbf{r}$.

Based on Equation 9, we can further derive the whole optimization objective $\mathcal{L}_{\text{cib}}$ as:

$$\begin{aligned} \mathcal{L}_{\text{cib}} &= \Big[ \mathcal{I}(\mathbf{u}; \mathbf{z}|\mathbf{r}) + \mathcal{I}(\mathbf{r}; \mathbf{k}|\mathbf{u}) \Big] - \beta \mathcal{I}(\mathbf{r}; \mathbf{z}) \\ &\leq \mathcal{D}_{\mathcal{KL}}\big(\mathcal{N}_\psi||\mathcal{N}_\eta\big) + \mathcal{D}_{\mathcal{KL}}\big(\mathcal{N}_\eta||\mathcal{N}_\psi\big) - \beta \mathcal{I}(\mathbf{k}; \mathbf{z}) \\ &= \mathcal{D}_{\mathcal{SKL}}(\mathcal{N}_\psi||\mathcal{N}_\eta) - \beta \mathcal{I}(\mathbf{k}; \mathbf{z}). \end{aligned} \tag{33}$$

By reparameterizing the objective, we can derive the final loss function:

$$\mathcal{L}_{\text{info}}(\psi, \eta) = \mathbb{E}_\mathcal{B} \Big[ -\mathcal{I}(\mathbf{z}; \mathbf{k}) + \alpha \mathcal{D}_{\mathcal{SKL}}\big(\mathcal{N}_\psi||\mathcal{N}_\eta\big) \Big]. \tag{34}$$

## C.4 PROOF OF THE ACTION VALUE BOUND

We provide the detailed proof of the Theorem 4.4.

**Theorem 4.4.** *Let $Z$ be a learned robust action-value representation from any observation-action pair $(o, a) \in \mathcal{O} \times \mathcal{A}$, $Q_e^\pi : \mathcal{O} \times \mathcal{A} \to \mathbb{R}$ be the true action-value function of a policy $\pi$ in the environment $e \in \mathcal{E}$, $f_e^* : \mathcal{Z} \to \mathbb{R}$ be the optimal linear mapping on the representation space, and $\epsilon$ be a bound of estimation error for each compressed reward representation $z$, i.e., $|f_e^*(z) - r| \leq \epsilon$. For any $(o, a) \in \mathcal{O} \times \mathcal{A}$ and $e \in \mathcal{E}$, we have*

$$0 \leq \big| Q_e^\pi(o, a) - f_e^*\big(Z(o, a)\big) \big| \leq \frac{1}{1 - \gamma}\epsilon.$$

*Proof.*

$$\begin{aligned} \big\| Q_e^\pi(o, a) - f_e^*\big(Z(o, a)\big) \big\|_1 &= \left\| \mathbb{E}_{\pi^*}^e \left[ \sum_{t=0}^\infty \gamma^t r_t \right] - \mathbb{E}_{\pi^*}^e \left[ \sum_{t=0}^\infty \gamma^t f_e^*(z) \right] \right\|_1 \\ &= \left\| \mathbb{E}_{\pi^*}^e \left[ \sum_{t=0}^\infty \gamma^t (r_t - f_e^*(z)) \right] \right\|_1 \\ &\leq \mathbb{E}_{\pi^*}^e \left[ \sum_{t=0}^\infty \gamma^t \|r_t - f_e^*(z)\|_1 \right] \\ &\leq \mathbb{E}_{\pi^*}^e \left[ \sum_{t=0}^\infty \gamma^t \cdot \epsilon \right] \\ &= \frac{1}{1 - \gamma}\epsilon. \end{aligned}$$

$\square$

## D  ARCHITECTURE AND PSEUDOCODE

As we directly use the embeddings from critic's center layer as robust action-value representations, we do not need to change the architecture of commonly used VRL algorithms. Specifically, we follow the traditional VRL actor-critic architecture to use a 3-layer feed-forward ConvNet with no residual connection as the encoder. Then, we apply a 3-layer MLP with hidden size 1024 as each critic, and we use Equation 15 to regulate the embeddings of the second layer of each critic. The actor uses the same architecture as the critic. Moreover, we use an additional 2-layer MLP with hidden size 1024 as the reward model $\psi$ to output the compressed reward representations $\mathbf{z}$, as well as $\eta$ to output the representations $\mathbf{k}$ from one-step rewards. As the inputs are image observations, the reward model $\psi$ is after the shared encoder for the actor and critic. We provide the architecture and the algorithm of ROUSER in Figure 1 and Algorithm 1, respectively.

---

**Algorithm 1** ROUSER in a general actor-critic framework

**Initialize** the critic network $Q_\theta$ with parameters $\theta$, actor network $\pi_\phi$ with parameters $\phi$, reward model $\psi$, and the linear mapping $f$.
**Initialize** target network $Q_{\bar{\theta}}$ with weight $\bar{\theta} \leftarrow \theta$.
**Initialize** the replay buffer $\mathcal{B}$.
**for** $t = 1, \ldots, T$ **do**
    **if** $t \leq T_0$ **then**
        Randomly select $a$ under $o$.
        Execute $a$ to obtain $r$ and $o'$.
    **else**
        $a \sim \pi_\phi(\cdot|o)$
        $o' \sim p(o'|o,a)$
        Sample a batch $\{o,a,r,o\prime\}$ from $\mathcal{B}$.
        **Update:**

$$f \leftarrow f - \lambda_f \nabla_f \mathcal{L}_{\text{pred}} \qquad\qquad \textit{linear network}$$
$$\psi \leftarrow \psi - \lambda_\psi \nabla_\psi \big(\mathcal{L}_{\text{pred}} + \mathcal{L}_{\text{info}}\big) \qquad\qquad \textit{reward network}$$
$$\theta \leftarrow \theta - \lambda_\theta \nabla_\theta (\mathcal{L}_{\text{robust}} + \mathcal{L}_{\text{critic}}) \qquad\qquad \textit{critic network}$$
$$\phi \leftarrow \phi - \lambda_\phi \nabla_\phi \mathcal{L}_{\text{actor}} \qquad\qquad \textit{actor network}$$
$$\bar{\theta} \leftarrow \tau\theta + (1-\tau)\bar{\theta} \qquad\qquad \textit{target network}$$

    **end if**
    $\mathcal{B} \leftarrow \mathcal{B} \cup \{o,a,r,o'\}$
**end for**

---

## E  EXPERIMENTS

### E.1  EXPERIMENTS ON DMC

**Environment Setting.** *Color Distractions.* On DMC benchmark, we apply the similar treatment of dynamic color distraction (Stone et al., 2021) to the objects of the environments. Specifically, we uniformly sample the color $x_0 \sim \mathcal{U}(x - \beta, x + \beta)$ for each channel at the start of each episode, where $x$ is the origin color in DCS, and $\beta$ is a hyperparameter. We leverage a dynamic setting where the color $x_t$ can change to $x_{t+1} = clip(\widehat{x}_{t+1}, x_t - \beta, x_t + \beta)$, where $\widehat{x}_{t+1} \sim \mathcal{N}(x_t, 0.03 \cdot \beta)$. We train the agents on the environment with weak dynamic color distractions ($\beta = 0.2$). Then, we evaluate all agents in the test environments with strong dynamic color distractions ($\beta = 0.5$).

*Background Distractions.* We adopt the dynamic background settings from Stone et al. (2021). To establish different training environments, we utilize $N$ videos from the DAVIS 2017 training set, where $N$ represents the number of training environments. Each environment uses one video as the background and randomly samples a scene and a frame from the video at the start of every episode. Additionally, we set $\beta_{\text{bg}} = 1.0$, indicating that we use the distracting background instead of the original skybox. For evaluation, we apply 30 videos from the DAVIS 2017 validation dataset as the unseen backgrounds. In each episode of the test environment, we randomly select one of the 30 dynamic backgrounds.

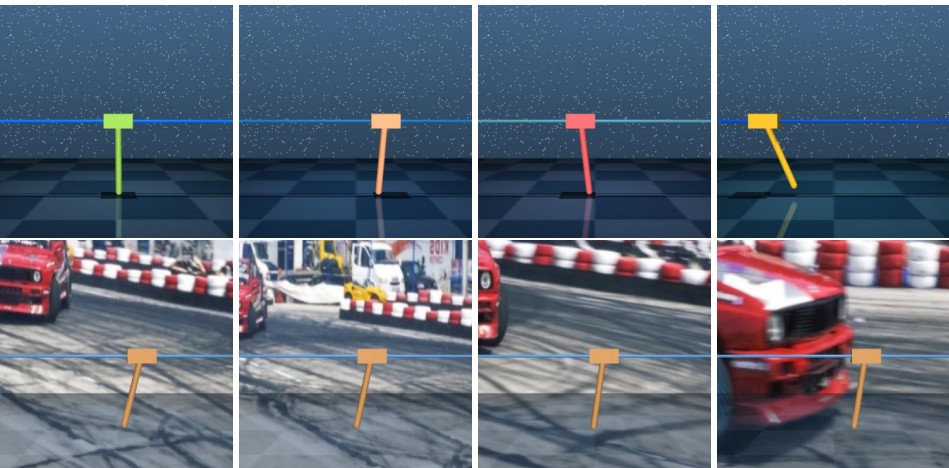

Figure 6: The dynamics color and background distractions using in Section 5.1. The first row illustrates the dynamic color changes, and the second row shows one of dynamic backgrounds.

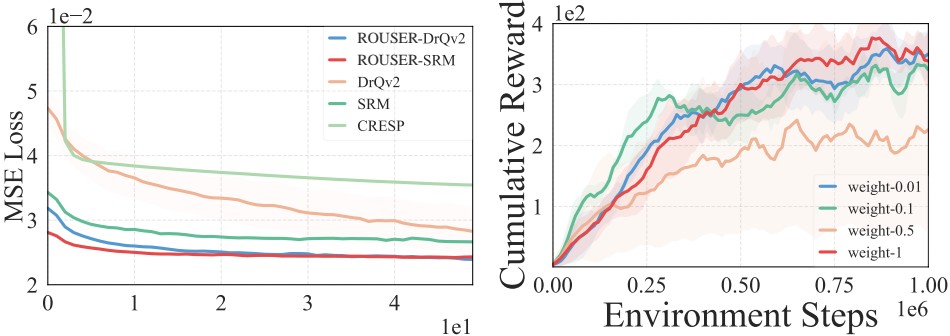

Figure 7: On cheetah_run task under color changes, the left of this figure shows the curves of MSE loss for capturing long-term information, and the right illustrates the performance of weight $c$.

**Network.** For the shared encoder, we use the default setting of baselines combined with our approach. For example, in ROUSER-SRM, we use a shared pixel encoder with 11 convolutional layers to extract the image information, and employ $3 \times 3$ kernels and 32 filters with a stride of 2 for the first convolutional layer and 1 for others. After each convolution, following the commonly used setting of VRL methods (e.g., DrQ, DrQv2, and SRM), we apply a fully connected layer to output 50-dimensional representations normalized by LayerNorm and a tanh activation. Note that the actor and critic each have a fully connected layer.

After the fully connected layer, the actor network is parameterized with 3 fully connected layers using ReLU activations up until the last layer. The critic network employs Double Q-learning technique, where each Q-value is learned using a 3-layer fully connected network, similar to the one used in the actor network. The output dimension of these hidden layers in the actor and the critic network is 1024. Moreover, the gradients of the shared pixel encoder are computed through the critic's optimizer rather than the actor's.

The reward model also has a fully connected layer to output 50-dimensional representations normalized by LayerNorm and a tanh activation from the outputs of the shared encoder. After the fully connected layer, the reward model has a 3-layer MLP. The output dimension of these hidden layers in the reward model is also 1024. The final outputs of the reward model is twice the dimension of compressed reward representations, half of which is the mean and half of which is the variance. Thus, we can use the outputs to model the Gaussian distribution of compressed reward representations. Then, we use a linear layer, mapping compressed reward representations into the corresponding one-step rewards. Moreover, we use a linear layer to get the representations $\mathbf{k}$ from one-step rewards.

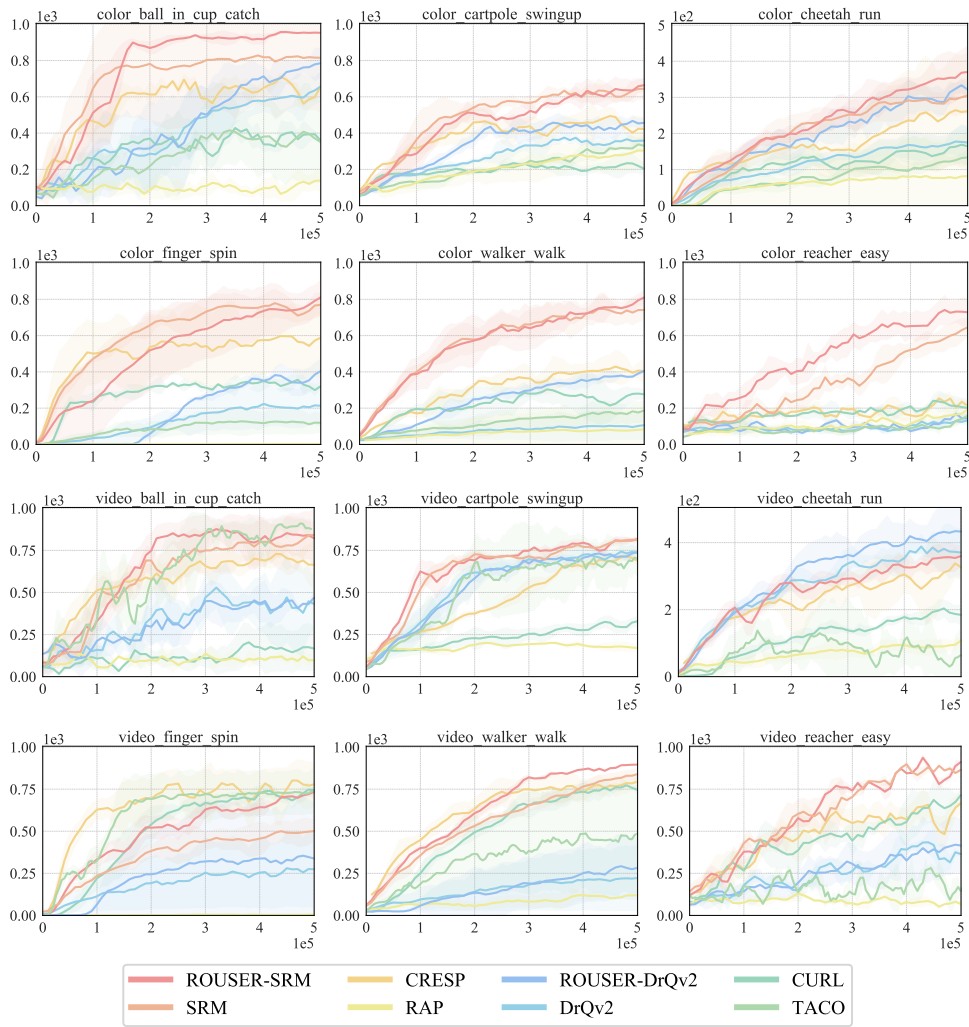

Figure 8: **Training curves** over six seeds on DMC benchmark with *color and video background distractions*. Each point from the curves is evaluated over ten episodes on test environments with unseen distractions.

**Implementations.** In the implementation of VRL, the multi-step reward is a commonly used trick without any theoretical guarantee: $r(o_t, a_t) = \sum_{i=0}^{N} \gamma^i \hat{r}(o_{t+i}, a_{t+i})$, where $\hat{r}(\cdot, \cdot)$ is the raw reward function. Many VRL methods (Yarats et al., 2022; Huang et al., 2022; Hessel et al., 2018) use $N = 3$. In our approach, we do not change the default setting when combined with baselines. It is worth noting that we change the mode of our reward model. Specifically, we map the outputs of reward model into the $N$ raw rewards instead of the multi-step reward, thus learning the compressed reward representations. Unlike CRESP, which predicts reward sequence distributions to update the shared encoder for representation learning, our reward model does not update the shared encoder with this mapping, although it establishes a mapping from the shared encoder's outputs to the multi-step rewards. This is because that we believe the multi-step rewards with length of $N = 3$ do not involve much long-term robust information for sequential decision-making. We demonstrate this in the left of Figure 7. The experiment setting for this figure is the same as in Figure 4 (a). We use 200K collected data to learn a 2-layer MLP projector that maps the fixed representations from CRESP and ROUSER-DrQv2 to the long-term sequential rewards with the length of 300. The curves demonstrate that ROUSER can effectively capture more long-term robust information than CRESP.

**Compute Resource.** We use NVIDIA GeForce RTX 3090 GPUs for six tasks on DMC benchmark under backgrounds or color changes. Trials of DrQv2, SRM, ROUSER-DrQv2, and ROUSER-SRM

Table 1: Means and standard errors on six DMC environments with *unseen dynamic color changes*. Each result is averaged over 100 episodes at 500K environment steps using six random seeds.

| ALGO | ball_in_cup_catch | cartpole_swingup | cheetah_run | finger_spin | walker_walk | reacher_easy |
|---|---|---|---|---|---|---|
| ROUSER-SRM | **942.6 ± 12.3** | **683.5 ± 37.8** | **370.0 ± 25.1** | **853.4 ± 25.6** | **814.0 ± 19.2** | **754.8 ± 39.7** |
| ROUSER-DrQv2 | 780.9 ± 60.3 | 461.4 ± 15.6 | **371.6 ± 23.7** | 424.2 ± 49.8 | 417.7 ± 22.7 | 154.6 ± 52.8 |
| SRM | 826.0 ± 125.6 | 640.2 ± 15.8 | 299.5 ± 23.4 | 779.7 ± 46.8 | 776.5 ± 27.8 | 659.4 ± 18.4 |
| DrQv2 | 632.1 ± 49.5 | 377.0 ± 24.7 | 140.2 ± 10.2 | 234.0 ± 54.9 | 115.3 ± 41.8 | 93.7 ± 20.4 |
| CRESP | 584.8 ± 57.4 | 410.6 ± 15.2 | 241.5 ± 87.3 | 574.0 ± 204.2 | 394.1 ± 65.0 | 270.9 ± 17.6 |
| CURL | 339.2 ± 51.5 | 205.5 ± 31.5 | 173.3 ± 46.6 | 279.2 ± 27.6 | 278.2 ± 46.3 | 198.2 ± 37.4 |
| TACO | 360.5 ± 192.8 | 385.9 ± 26.3 | 138.7 ± 17.2 | 130.8 ± 45.7 | 177.8 ± 31.8 | 122.9 ± 28.2 |
| RAP | 149.2 ± 34.2 | 304.4 ± 44.4 | 90.8 ± 57.6 | 1.5 ± 0.5 | 86.5 ± 62.5 | 209.8 ± 59.7 |

Table 2: Means and standard errors on six DMC environments with *unseen dynamic backgrounds*. Each result is averaged over 100 episodes at 500K environment steps using six random seeds.

| ALGO | ball_in_cup_catch | cartpole_swingup | finger_spin | cheetah_run | walker_walk | reacher_easy |
|---|---|---|---|---|---|---|
| ROUSER-SRM | **868.9 ± 47.0** | **822.4 ± 10.8** | **776.8 ± 24.0** | 372.9 ± 10.8 | **894.0 ± 12.6** | **901.5 ± 18.2** |
| ROUSER-DrQv2 | 523.1 ± 67.5 | 731.9 ± 10.5 | 348.2 ± 174.1 | **408.0 ± 22.1** | 358.6 ± 74.7 | 440.0 ± 70.9 |
| SRM | 818.1 ± 47.6 | 807.7 ± 14.5 | 527.5 ± 27.2 | 333.8 ± 22.9 | 823.1 ± 17.5 | 889.2 ± 14.2 |
| DrQv2 | 474.3 ± 72.6 | 717.8 ± 15.0 | 309.7 ± 113.8 | 359.2 ± 19.2 | 237.7 ± 106.9 | 404.5 ± 32.9 |
| CRESP | 665.3 ± 75.5 | 689.8 ± 20.0 | **778.1 ± 62.9** | 327.4 ± 20.1 | 794.1 ± 33.9 | 667.7 ± 33.5 |
| CURL | 167.0 ± 57.4 | 329.6 ± 18.4 | 745.9 ± 31.8 | 185.0 ± 15.9 | 746.1 ± 16.7 | 714.9 ± 33.1 |
| TACO | 816.1 ± 42.6 | 726.6 ± 67.4 | 737.2 ± 99.4 | 64.3 ± 59.5 | 505.4 ± 151.2 | 198.8 ± 51.2 |
| RAP | 76.1 ± 14.6 | 170.8 ± 21.7 | 6.9 ± 4.0 | 96.7 ± 12.4 | 115.1 ± 5.7 | 150.9 ± 16.4 |

on *reacher_easy* task from DMC under dynamic backgrounds are trained for 12.24, 10.83, 16.93, and 15.83 hours on average. Moreover, the agents of ROUSER-DrQv2 and ROUSER-SRM require approximately 2800MB and 3600MB of memory using the batch size of 256, respectively.

**Results.** We compare our approach ROUSER against several SOTA method: (1) CURL (Laskin et al., 2020), which leverages contrastive learning to maximize the mutual information between representations from observations and augmentations. (2) DrQv2 (Yarats et al., 2022), which is the prior state-of-the-art DRL algorithm for sample efficiency. (3) SRM (Huang et al., 2022), which adopts augmentation in the frequency domain to facilitate the learning of robust policies. (4) RAP (Chen & Pan, 2022), which effectively enhances the robustness of representations by leveraging the behavioral similarity. (5) CRESP (Yang et al., 2022), which predicts the characteristic function of reward sequences to learn task-relevant representations for generalization. (6) TACO (Zheng et al., 2024), which learns state and action representations that encompass sufficient information for control to improve sample efficiency. Moreover, we illustrate image observations from the environments with dynamic color and background distractions in Figure 6.

We provide the means and standard errors at 500K environment steps in Tables 1 and 2. Note that Table 1 lists the detailed results of Figure 2 under unseen dynamic color changes, and Table 2 lists the detailed results of Figure 2 under unseen dynamic video backgrounds. As shown in Tables 1 and 2, ROUSER outperforms other baselines in 11 out of 12 experiments. Even in finger_spin task under dynamic backgrounds, our approach ROUSER is almost on par with the highest performance. In addition, we illustrate the training curves over six seeds in Figure 8.

**Analysis of learned Representations for Capturing Long-Term Robust Information.** In Section 5.3, we present results from a task where learned representations are used to predict the average of future reward sequences over a length of 300. These results indicate the effectiveness of ROUSER in capturing long-term information.

Moreover, we conduct additional experiments with varying reward sequence lengths, including 50, 100, 300, and 500. We report the results in Table 3, which are averaged over three seeds on cheetah_run task with unseen color distractions. These results, averaged over three seeds on cheetah_run task with unseen color distractions, demonstrate that the representations learned by ROUSER are able to accurately predict reward sequences of different lengths, with performance improving as the reward sequences become longer.

| Reward Sequence | ROUSER-SRM | SRM | ROUSER-DrQv2 | DrQv2 |
|---|---|---|---|---|
| Length=50 | **0.0169 ± 0.0051** | 0.0170 ± 0.0056 | **0.0169 ± 0.0053** | 0.0173 ± 0.0029 |
| Length=100 | 0.0213 ± 0.0070 | 0.0210 ± 0.0069 | **0.0205 ± 0.0065** | 0.0211 ± 0.0004 |
| Length=300 | 0.0244 ± 0.0122 | 0.0268 ± 0.0179 | **0.0239 ± 0.0182** | 0.0283 ± 0.0029 |
| Length=500 | 0.0294 ± 0.0091 | 0.0383 ± 0.0004 | **0.0287 ± 0.0087** | 0.0319 ± 0.0011 |

Table 3: MSE results and standard deviations with different reward sequence lengths over three seeds, including 50, 100, 300, and 500.

| Game | PPO | MixReg | QR-DQN | ROUSER-QRDQN |
|---|---|---|---|---|
| bigfish | 3.7 ± 1.3 | 7.1 ± 1.6 | 12.2 ± 1.1 | **13.6 ± 5.2** |
| caveflyer | 5.1 ± 0.4 | 6.1 ± 0.6 | 6.0 ± 0.7 | **6.4 ± 0.4** |
| dodgeball | 1.6 ± 0.1 | 1.7 ± 0.4 | 7.2 ± 0.2 | **8.0 ± 0.4** |

Table 4: Average Procgen scores on test levels after training on 25M environment steps. The mean and standard deviation are computed using five results with different random seeds. We boldface the results that have highest means. This table corresponds to the results in Figure 3.

### E.2 EXPERIMENTS ON PROCGEN

**Settings.** Procgen benchmark consists of 16 procedurally generated games. Each of these games has procedurally generated levels which present agents with meaningful generalization challenges (Raileanu et al., 2021). All environments of these games use a discrete 15-dimensional action space, and produce $64 \times 64 \times 3$ RGB observations. Following Raileanu et al. (2021), we use the easy setting, where agents are learned on the training environments with 200 levels and tested on environments with unseen levels.

**Implementations.** Value-based VRL methods mainly focus on non-continuous control tasks. They often use a critic to estimate the action-value function without a actor/policy network. Thus, their critic inputs an image observation and outputs a vector of action values for all possible discrete actions. Note that the last layer of their critic maps the representations of image observations into the action values of all possible discrete actions. Therefore, in the combination of ROUSER and QR-DQN, for the action value of each possible discrete action, we replace its last layer mapping into a 2-layer ensemble MLP, where the ensemble size is the number of discrete actions. The inputs of this 2-layer ensemble MLP are robust action-value representations, and this MLP outputs the quantile values of the action-value function for each discrete action.

Moreover, we use the same design for the reward model $\psi$ with a 3-layer MLP. Specifically, the reward model uses the image observations as inputs and outputs the compressed reward representations for each discrete action. It uses actions from a batch to select compressed representations for updating.

**Results.** We compare our approach with: (1) PPO (Schulman et al., 2017), a popular policy gradient baseline upon which many competitive methods are developed; (2) MixReg (Wang et al., 2020), a regularization method applicable to both policy gradient and value-based RL algorithms to enhance generalization on Procgen; (3) QR-DQN, a prior SOTA value-based method on Procgen.

Table 4 lists the Procgen scores on test levels after training on 25M environment steps using the batch size of 256. The mean and standard deviation are averaged over five runs with different random seeds. We adapt the tables from Raileanu & Fergus (2021). These results demonstrate that ROUSER can be also applicable to value-based RL algorithms, outperforming several previous methods.

### E.3 EXPERIMENTS ON MUJOCO

To demonstrate that ROUSER can not only improve robustness in VRL but also enhance sample efficiency in traditional state-based RL (i.e., RL using a vector as a state), we further conduct experiments on MuJoCo (Todorov et al., 2012) using vector states. We combine our approach with DDPG (Lillicrap et al., 2016) and SAC (Haarnoja et al., 2018), and further compare it against the

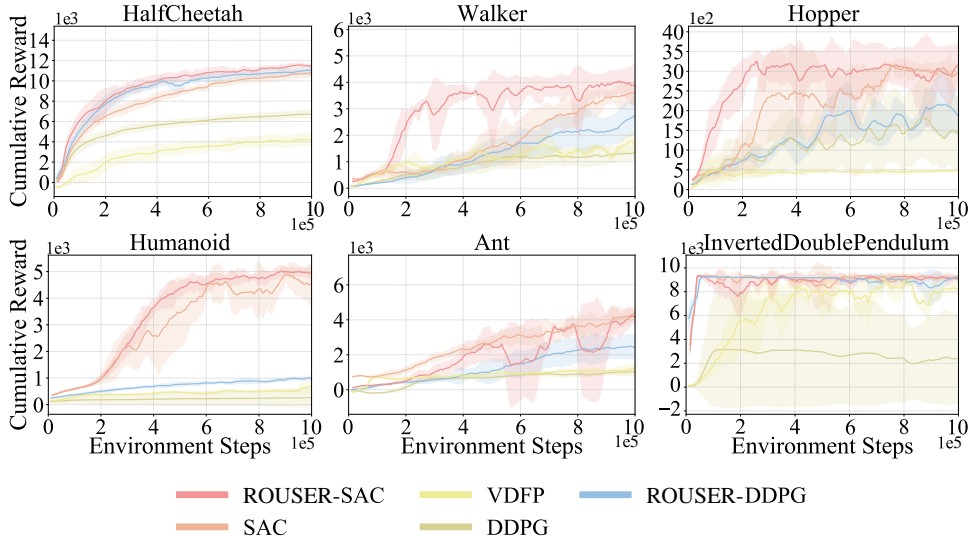

Figure 9: Training curves during the training process with 1M steps over six random seeds.

Table 5: Hyperparameters of ROUSER on DMC and Procgen benchmarks.

| Parameter | DMC Value | Procgen Value |
|---|---|---|
| Optimizer | Adam | Adam |
| learning rate of critic/action-value network | $1 \cdot 10^{-4}$ | $2.5 \cdot 10^{-4}$ |
| learning rate of robust action-value representations | $1 \cdot 10^{-5}$ | $2.5 \cdot 10^{-5}$ |
| learning rate of actor network | $1 \cdot 10^{-4}$ | None |
| learning rate of reward model | $1 \cdot 10^{-4}$ | $2.5 \cdot 10^{-4}$ |
| target smoothing coefficient | 0.01 | 0.005 |
| update frequency of target network | 2 | 1 |
| dimension of robust action-value representations $D$ | 512 | 512 |
| batch size | 256 | 512 |
| total environment steps $T$ | $10^6$ | $2.5 \cdot 10^7$ |
| $\alpha$ scheduler start value | $10^{-3}$ | $10^{-2}$ |
| $\alpha$ scheduler end value | $10^{-2}$ | $10^{-1}$ |
| reward model update times for each step | 5 | 5 |
| nstep | 3 | 3 |

SOTA of task decomposition methods, VDFP (Tang et al., 2021). We illustrate training curves (mean and standard deviation) on six tasks from MoJoCo during the training process in Figure 9. Each result is run for 1 million time steps over six random seeds. Figure 9 shows that ROUSER outperforms baselines, achieving an average improvement of **+18.7%**.

### E.4    EXPERIMENTS ON DMC-GB

**Environment Setting.**    To comprehensively evaluate our approach, we have conducted additional experiments on six DMC-GB (Hansen & Wang, 2021) tasks across all their settings (i.e., color_easy, color_hard, video_easy, and video_hard). All these experiment settings are the same as Hansen & Wang (2021) and Hansen et al. (2021). Moreover, we compare our approach against several SOTA generalization VRL methods, including PIE-G (Yuan et al., 2022), SVEA (Hansen et al., 2021), DrG (Ha et al., 2023), SECANT (Fan et al., 2021), SGQN (Bertoin et al., 2022), and MIIR (Wang et al., 2024b). Most of the baseline results we report are taken directly from their respective papers. For cases where results are not provided, we ran experiments using the available source code and hyperparameters under three random seeds.

Table 6: Hyperparameters of ROUSER on MuJoCo benchmark for sample efficiency.

| Parameter | Value |
|---|---|
| Optimizer | Adam |
| learning rate of critic network | $1 \cdot 10^{-3}$ |
| learning rate of robust action-value representations | $1 \cdot 10^{-4}$ |
| learning rate of actor network | $1 \cdot 10^{-4}$ |
| learning rate of reward model | $5 \cdot 10^{-5}$ |
| target smoothing coefficient | 0.01 |
| update frequency of target network | 2 |
| dimension of robust action-value representations $D$ | 512 |
| batch size | 256 |
| total environment steps $T$ | $10^6$ |
| $\alpha$ scheduler start value | $10^{-3}$ |
| $\alpha$ scheduler end value | $10^{-2}$ |
| nstep | 1 |

Table 7: Results and standard deviations on DMC-GB under the color_easy and color_hard settings. Results of our approach are averaged over three random seeds.

| color_easy | ROUSER-SRM | PIE-G | DrG | SVEA | SGQN | TIA | SECANT |
|---|---|---|---|---|---|---|---|
| ball_in_cup_catch | **963** $\pm$ **8** | $955 \pm 4$ | $831 \pm 92$ | $959 \pm 2$ | $907 \pm 71$ | $652 \pm 402$ | - |
| cartpole_swingup | **841** $\pm$ **20** | $624 \pm 52$ | $701 \pm 43$ | $826 \pm 20$ | $598 \pm 92$ | $483 \pm 62$ | - |
| cheetah_run | **616** $\pm$ **13** | $429 \pm 12$ | $375 \pm 31$ | $587 \pm 39$ | $304 \pm 43$ | $281 \pm 2$ | - |
| finger_spin | **933** $\pm$ **11** | $845 \pm 5$ | $876 \pm 79$ | $892 \pm 59$ | $628 \pm 46$ | $410 \pm 49$ | - |
| walker_walk | **942** $\pm$ **6** | $909 \pm 40$ | $812 \pm 33$ | $907 \pm 23$ | $845 \pm 26$ | $780 \pm 140$ | - |
| walker_stand | **971** $\pm$ **2** | $968 \pm 4$ | $910 \pm 20$ | $965 \pm 10$ | $963 \pm 11$ | $895 \pm 39$ | - |
| **color_hard** | ROUSER-SRM | PIE-G | DrG | SVEA | SGQN | TIA | SECANT |
| ball_in_cup_catch | **964** $\pm$ **5** | $960 \pm 3$ | $607 \pm 46$ | $961 \pm 7$ | $905 \pm 71$ | $575 \pm 375$ | $958 \pm 7$ |
| cartpole_swingup | $799 \pm 24$ | $520 \pm 69$ | $523 \pm 38$ | $837 \pm 23$ | $540 \pm 76$ | $407 \pm 173$ | **866** $\pm$ **15** |
| cheetah_run | $562 \pm 10$ | $376 \pm 27$ | $219 \pm 8$ | $456 \pm 62$ | $277 \pm 43$ | $249 \pm 25$ | **582** $\pm$ **64** |
| finger_spin | **979** $\pm$ **9** | $838 \pm 10$ | $758 \pm 124$ | $977 \pm 5$ | $461 \pm 5$ | $408 \pm 79$ | $910 \pm 115$ |
| walker_walk | **916** $\pm$ **9** | $824 \pm 92$ | $725 \pm 134$ | $760 \pm 145$ | $692 \pm 153$ | $616 \pm 243$ | $856 \pm 31$ |
| walker_stand | **962** $\pm$ **2** | $948 \pm 15$ | $731 \pm 21$ | $942 \pm 26$ | $905 \pm 34$ | $841 \pm 37$ | $939 \pm 7$ |

1. We run PIE-G, DrG, and SGQN on all DMC-GB tasks across color_easy and color_hard settings.

2. We run DrG on finger_spin and walker_stand tasks across video_easy and video_hard settings.

3. We run SVEA on all DMC-GB tasks in the color_easy setting and on cheetah_run task in the color_hard setting.

4. We run TIA [2] on all DMC-GB tasks in all four settings.

Note that since SECANT and MIIR did not release their source code, we only include the results reported in their respective papers.

**Results.** We provide all results in Tables 7 and 8. These results indicate that ROUSER outperforms the aforementioned baselines in 17 out of 24 settings, demonstrating ROUSER's robustness and effectiveness in DMC-GB tasks.

E.5 HYPERPARAMETERS

**Hyperparameters.** We provide the hyperparameters for DMC, Procgen, and MuJoCo in Tables 5 and 6. In DMC benchmark, the hyperparameter $\alpha$ in the loss function $\mathcal{L}_{\text{info}}$ can be tuned from a small value $10^{-3}$ to $10^{-2}$ by an exponential scheduler.

---

[2]We use the reproduced version of TIA provided at https://github.com/zchuning/repo.

Table 8: Results and standard deviations on DMC-GB under the video_easy and video_hard settings. Results of our approach are averaged over three random seeds.

| video_easy | ROUSER | PIE-G | DrG | SVEA | SGQN | MIIR | TIA | SECANT |
|---|---|---|---|---|---|---|---|---|
| ball_in_cup_catch | **979 ± 21** | 922 ± 20 | 701 ± 36 | 871 ± 22 | 950 ± 24 | 973 ± 2 | 664 ± 416 | 903 ± 49 |
| cartpole_swingup | 830 ± 21 | 587 ± 61 | 572 ± 25 | 782 ± 27 | 761 ± 28 | **858 ± 16** | 326 ± 139 | 752 ± 38 |
| cheetah_run | 397 ± 30 | 287 ± 20 | 547 ± 21 | 249 ± 20 | 308 ± 34 | 393 ± 57 | 225 ± 41 | **428 ± 70** |
| finger_spin | 970 ± 6 | 837 ± 107 | 751 ± 43 | 808 ± 33 | 956 ± 26 | **978 ± 9** | 295 ± 16 | 861 ± 102 |
| walker_walk | **925 ± 17** | 871 ± 22 | 902 ± 23 | 819 ± 71 | 910 ± 24 | 919 ± 30 | 696 ± 245 | 842 ± 47 |
| walker_stand | **973 ± 3** | 957 ± 12 | 910 ± 17 | 961 ± 8 | 955 ± 9 | 971 ± 3 | 903 ± 18 | 932 ± 15 |
| **video_hard** | ROUSER | PIE-G | DrG | SVEA | SGQN | MIIR | TIA | SECANT |
| ball_in_cup_catch | **940 ± 5** | 786 ± 47 | 635 ± 26 | 403 ± 174 | 782 ± 57 | 929 ± 9 | 314 ± 36 | - |
| cartpole_swingup | **769 ± 37** | 401 ± 21 | 545 ± 23 | 393 ± 45 | 544 ± 43 | 765 ± 22 | 189 ± 29 | - |
| cheetah_run | 270 ± 12 | 154 ± 17 | **489 ± 11** | 105 ± 37 | 135 ± 44 | 268 ± 73 | 76 ± 25 | - |
| finger_spin | 937 ± 10 | 762 ± 59 | 437 ± 61 | 335 ± 58 | 822 ± 24 | **956 ± 18** | 182 ± 55 | - |
| walker_walk | **868 ± 59** | 600 ± 28 | 782 ± 37 | 377 ± 93 | 739 ± 21 | 821 ± 58 | 166 ± 60 | - |
| walker_stand | **968 ± 4** | 852 ± 56 | 819 ± 58 | 834 ± 46 | 851 ± 24 | 965 ± 2 | 673 ± 142 | - |

**Hyperparameter Search.** In the right of Figure 7, we illustrate the results (mean and standard deviation) of ROUSER-DrQv2 on cheetah_run task from DMC benchmark under color changes for searching the learning rate $l$ of robust action-value representations. Based on the learning rate $\hat{l} = 0.0001$, we introduce the weight $c$ such that $l = c \cdot \hat{l}$. Each curve is averaged over three seeds.

Moreover, in Figure 5 of the main text, we report the results without standard errors of different $\alpha$ with three random seeds. Here, we provide the detailed results (refer to Table 9).

Table 9: The mean and standard error of the results in Figure 5.

| $\alpha$ | scale-0.3 | scale-0.4 | scale-0.5 | scale-0.6 |
|---|---|---|---|---|
| 0.01-0.1 | 649.7 ± 9.1 | 488.0 ± 15.7 | 363.0 ± 17.3 | **316.1 ± 9.1** |
| 0.001-0.01 | 663.7 ± 7.5 | **524.4 ± 14.5** | **398.4 ± 19.4** | 283.3 ± 11.2 |
| 0.0001-0.001 | **683.1 ± 2.2** | 510.3 ± 16.8 | 367.9 ± 12.5 | 251.4 ± 11.5 |

