# OpenReview forum: "Learning Robust Representations with Long-Term Information for Generalization in Visual Reinforcement Learning"
_ICLR.cc/2025/Conference — ICLR 2025 Poster_

### Official Review · Reviewer_6q1B · 2024-10-30

**Soundness:** 3
**Presentation:** 3
**Contribution:** 3
**Rating:** 6
**Confidence:** 3

**Summary:**

This paper proposes a novel way for learning robust representations in image-based RL by learning through the proposed robust action-value representation using an information bottleneck framework. To address the challenge of unknown true action-values, this approach learns action-values indirectly through reward representations. The proposed method is validated in both continuous and discrete control domain. Experiments demonstrate improved performance over other image-based RL baselines on tasks with video background and object color changes.

**Strengths:**

- The paper is clearly written and easy to follow.
- The motivation and explanation for learning robust action-value representations, and learning robust action-value representations via reward representations are presented clearly.
- The experiment is conducted on both continuous and discrete control tasks.
- Applying ROUSER on top of DrQv2 and SRM shows consistent improvement.

**Weaknesses:**

- Lack of training curves on Distracting DMControl benchmark for Figures 1 and 2: Including training curves would help readers understand the sample efficiency and stability of the proposed method.
- In Figures 1 and 2, the baselines include both standard methods for image-based RL (e.g., DrQv2, CURL, TACO) and methods designed to be robust against distractors (e.g., SRM, CRESP, RAP). It would be better to reorganize the results, such as grouping methods by type or using different visual indicators for a clearer comparison.

Minor:
- Typo in line 430: Nore -> Note
- In Figure 4, axis labels should be included directly on the plots rather than in the caption.

**Questions:**

- In Section 5.3, the authors analyze the effectiveness of ROUSER in capturing long-term information with a fixed length of 300, It would be helpful to see how the effectiveness changes with varying lengths (e.g., 50, 100, 500).
- The recent method MIIR [1] also considers learning robust representations using the information bottleneck framework and conducts experiments on the Distracting DMControl benchmark. Could the author discuss the differences between ROUSER and MIIR, such as their theoretical motivations and empirical performance comparisons?

[1] Wang S, Wu Z, Wang J, Hu X, Lin Y, Lv K. How to Learn Domain-Invariant Representations for Visual Reinforcement Learning: An Information-Theoretical Perspective.

---

> ### Author Response · Authors · 2024-11-21
>
> We appreciate reviewer's insightful comments. We respond to each comment as follows and accordingly revise our manuscript, with all updates highlighted in *blue* for your convenience. We sincerely hope that our responses could properly address your concerns. If so, we would deeply appreciate it if you could raise your score. If not, please let us know your further concerns, and we will continue responding to the comments and enhancing our submission.
>
> - **Q1**. Lack of training curves on Distracting DMControl benchmark for Figures 1 and 2: Including training curves would help readers understand the sample efficiency and stability of the proposed method.
>   - **A1**. Thank you for pointing this out. We have included **all training curves** on Distracting DMControl benchmark in Figure 8 of our revision (see *Lines 1188-1223*).
> - **Q2**. In Figures 1 and 2, the baselines include both standard methods for image-based RL (e.g., DrQv2, CURL, TACO) and methods designed to be robust against distractors (e.g., SRM, CRESP, RAP). It would be better to reorganize the results, such as grouping methods by type or using different visual indicators for a clearer comparison.
>   - **A2**. Thanks for your valuable suggestions.
>     - In Lines 432-453 of our revision, we updated Figure 2 (a merge of Figures 1 and 2 from our previous manuscript) by **grouping** methods based on their **type**. This improves clarity, making the comparisons across methods more intuitive.
>       - We used **warm colors** to show VRL methods designed to improve robustness against distractions (i.e., SRM, CRESP, and RAP) on the left of each subplot. These methods were compared with ROUSER-SRM, which outperforms them in **11 out of 12** experiments.
>       - We used **cooler colors** to present standard VRL methods (i.e., DrQv2, CURL, and TACO) on the right of each subplot. We compare these methods with ROUSER-DrQv2, which outperforms DrQv2 in **all**  experiments and surpasses the other methods in **7 out of 12** experiments.
> - **Q3**. Typo in line 430: Nore -> Note
>   - **A3**. Thank you! We have corrected this typo in Line 431 of our revision.
> - **Q4**. In Figure 4, axis labels should be included directly on the plots rather than in the caption.
>   - **A4**. Thanks for bringing this to our attention. We have updated each subplot in Figure 4 to include x- and y-axis labels directly on the plots (see Lines 486-495), instead of referencing them in the caption.
> - **Q5**. In Section 5.3, the authors analyze the effectiveness of ROUSER in capturing long-term information with a fixed length of 300. It would be helpful to see how the effectiveness changes with varying lengths (e.g., 50, 100, 500).
>   - **A5**. Thanks for the kind and insightful comment.
>     - We have conducted additional experiments with varying reward sequence lengths, including 50, 100, 300, and 500. We reported the results in Table C1 below (also see Table 3 with its description in Lines 1287-1304 in Appendix E.1 of our revision).
>       These results, averaged over three seeds on cheetah_run task with unseen color distractions, demonstrate that the representations learned by ROUSER are able to **accurately** predict reward sequences of different lengths, with **performance improving** as the reward sequences become **longer**.
> ---
>   **Table C1**. MSE results and standard deviations with different reward sequence lengths over three seeds, including 50, 100, 300, and 500.
>   | Reward Sequence Length | ROUSER-SRM              | SRM                 | ROUSER-DrQv2            | DrQv2               |
>   | ---------------------- | ----------------------- | ------------------- | ----------------------- | ------------------- |
>   | 50                     | **0.0169 $\pm$ 0.0051** | 0.0170 $\pm$ 0.0056 | **0.0169 $\pm$ 0.0053** | 0.0173 $\pm$ 0.0029 |
>   | 100                    | 0.0213 $\pm$ 0.0070     | 0.0210 $\pm$ 0.0069 | **0.0205 $\pm$ 0.0065** | 0.0211 $\pm$ 0.0004 |
>   | 300                    | 0.0244 $\pm$ 0.0122     | 0.0268 $\pm$ 0.0179 | **0.0239 $\pm$ 0.0182** | 0.0283 $\pm$ 0.0029 |
>   | 500                    | 0.0294 $\pm$ 0.0091     | 0.0383 $\pm$ 0.0004 | **0.0287 $\pm$ 0.0087** | 0.0319 $\pm$ 0.0011 |

---

> ### Author Response · Authors · 2024-11-21
>
> - **Q6**. The recent method MIIR [1] also considers learning robust representations using the information bottleneck framework and conducts experiments on the Distracting DMControl benchmark. Could the author discuss the differences between ROUSER and MIIR, such as their theoretical motivations and empirical performance comparisons?
>   - **A6**. Thanks for bringing this to our attention. We summarize the major differences between ROUSER and MIIR as follows.
>     1. **Theoretical Analysis**. ROUSER provides a theoretical guarantee, establishing an **upper bound** between the true action-value function and the action-value function on top of learned representations (see Theorem 4.4 in our main text). In contrast, MIIR only derives its optimization objectives without offering any guarantees.
>     2. **Applicability*. We integrate ROUSER with both policy gradient and value-based VRL algorithms to estimate the action values, making it **suitable** for **continuous** and **discrete** control tasks. MIIR, however, has only been implemented and evaluated on continuous control tasks.
>     3. **Empirical Performance**.  For comparison with MIIR, we conducted additional experiments on six DMC-GB tasks under `video_easy` and `video_hard` settings, as MIIR did not release its source code and only reported results for these two settings. We report the average results and standard deviations over three seeds in Tables C2 and C3 below. These results demonstrate that ROUSER outperforms MIIR in **9 out of 12** experiments.
>        - We have included these results of Tables C2 and C3 in *Table 8* (Lines 1467–1480) of Appendix E.4 of our revision. For your convenience, we have highlighted these updates in blue.
>
>   [1] Wang S, Wu Z, Wang J, Hu X, Lin Y, Lv K. How to Learn Domain-Invariant Representations for Visual Reinforcement Learning: An Information-Theoretical Perspective.
> ---
>
>   **Table C2**. Results and standard deviations on DMC-GB under the `video_easy` setting. Results of our approach are over three random seeds.
>   |       | ball_in_cup_catch | cartpole_swingup | cheetah_run      | finger_spin      | walker_walk      | walker_stand    |
>   | --------------------- | ----------------- | ---------------- | ---------------- | ---------------- | ---------------- | --------------- |
>   | **ROUSER**            | **979 $\pm$ 21**  | 830 $\pm$ 21     | **397 $\pm$ 30** | 970 $\pm$ 6      | **925 $\pm$ 17** | **973 $\pm$ 3** |
>   | **MIIR**              | 973 $\pm$ 2       | **858 $\pm$ 16** | 393 $\pm$ 57     | **978 $\pm$ 9**  | 919 $\pm$ 30     | 971 $\pm$ 3  |
>
> ---
>
>   **Table C3**. Results and standard deviations on DMC-GB under the `video_hard` setting. Results of our approach are over three random seeds.
>   |   | ball_in_cup_catch | cartpole_swingup | cheetah_run      | finger_spin      | walker_walk      | walker_stand |
>   | --------------------- | ----------------- | ---------------- | ---------------- | ---------------- | ---------------- | --------------- |
>   | **ROUSER**            | **940 $\pm$ 5**   | **769 $\pm$ 37** | **270 $\pm$ 12** | 937 $\pm$ 10     | **868 $\pm$ 59** | **968 $\pm$ 4** |
>   | **MIIR**              | 929 $\pm$ 9       | 765 $\pm$ 22     | 268 $\pm$ 73     | **956 $\pm$ 18** | 821 $\pm$ 58     | 965 $\pm$ 2     |

---

> > ### Author Response · Authors · 2024-11-23
> > **Eagerly await your valuable feedback.**
> >
> > Dear Reviewer 6q1B,
> >
> > We deeply appreciate your valuable feedback and the time you've taken to review our work, especially during this busy period.
> >
> > We are reaching out to kindly inquire about the current status of your review regarding our submission. We sincerely hope that our responses have adequately addressed your concerns. Furthermore, we are eager to address any additional queries you might have, which will enable us to enhance our work further.
> >
> > Once again, thank you for your guidance and support.
> >
> > Best,
> > Authors

---

> > > ### Comment · Reviewer_6q1B · 2024-11-26
> > >
> > > Thanks the authors for addressing my questions and providing the new results. I will maintain my rating of 6.

---

> > > > ### Author Response · Authors · 2024-11-26
> > > > **Thanks for your kind support.**
> > > >
> > > > Dear Reviewer 6q1B,
> > > >
> > > > Thank you for taking the time to review our work! Would you kindly share any remaining concerns you may have? We will actively address and respond to your valuable comments.
> > > >
> > > > Best,
> > > >
> > > > Authors

---

### Official Review · Reviewer_uw2P · 2024-11-03

**Soundness:** 3
**Presentation:** 2
**Contribution:** 3
**Rating:** 8
**Confidence:** 3

**Summary:**

The authors propose a novel temporal difference objective (ROUSE) for learning representations that capture long-term information are robust to distractions and perturbations for visual reinforcement learning. To make the proposed method robust, the authors utilize the information bottleneck principle, and make sure the learned representations of state action pairs captures the information about the reward, while minimizing the amount of information it captures about the state itself.

ROUSE is can be integrated with a variety of critic-based methods. The authors integrated ROUSE with DrQ, SRM and QRDQN, and test on DeepMind Control, MuJoCo, ProcGen, showing improved robustness compared to the baselines.

**Strengths:**

- ROUSE is general can be applied to any visual RL algorithm that learns a critic;
- The proposed information-bottleneck objective is well-motivated mathematically. The authors provide detailed mathematical derivations of the objectives;
- The method shows good results on the proposed benchmarks and improves robustness to color change and background video distractors;

**Weaknesses:**

- The paper is not clearly written, it took me some time to go through and understand it. I suggest improving the flow a bit. For example, something that could help is better structure. In the beginning of section 4, I would say something about the big picture of the paper, as "We propose robust reward representations, then use td learning to learn robust value representations. Throughout, we use IB-based objective make representations robust." At the end of section 4, it would also help to list the objectives, or add the figure or pseudocode from the appendix. I understand you don't have much space, maybe decreasing the size of figures 1 and 2, or aggregating results over tasks could be a way to go here.

**Questions:**

Line 506: What information are you predicting in this frozen evaluation? Is it average of 300 rewards in the future? What exactly is the MSE between?


##### Writing

- 227: "one of action-value representations" : unclear wording, what does "one of" mean here?
- 270: we support that a compressed reward representation... :
- 299: model a reward model -> introduce a reward model
- 364: supports -> allows
- Sections 5.2 and 5.3 titles are not very informative. Applicability and effectiveness? what does that mean? Something more concrete would be better, like "Extending ROUSER to tasks with discrete actions" and "Analyzing learned representations"
- Figure 4: please put the x and y axis labels in the figure itself, at least in small font, that would make it easier to read.-
- 466: Reword to something like "In this subsection, motivated by the result in theorem 4.4, we extend RoUSER to..."
- 502: all methods' prediction: what does that mean?
- 504: "use a 2 layer mlp with a learning rate of 1e4". I understand what you're saying, but the sentence is awkward. I'd say "we train a 2 layer mlp on top of frozen..., and use 1e-4 learning rate to train it."
- 515: by searchings alpha... : reword this, this is not grammatically correct
- 539: "sheds light" -- instead say "we believe this method is general and applicable to other problems, such as self-driving"

---

> ### Author Response · Authors · 2024-11-21
>
> We appreciate the reviewer's insightful and constructive comments and suggestions, which help us significantly improve the clarity and quality of our manuscript. We have carefully addressed these concerns and accordingly revised the manuscript, with all updates highlighted in **blue** for your convenience. We sincerely hope that our responses could properly address your concerns. If so, we would deeply appreciate it if you could raise your score. If not, please let us know your further concerns, and we will continue actively responding to the comments and enhancing our submission.
>
> - **Q1**. The paper is not clearly written, it took me some time to go through and understand it. I suggest improving the flow a bit. For example:
>
>   (1) something that could help is better structure. In the beginning of section 4, I would say something about the big picture of the paper, as "We propose robust reward representations, then use td learning to learn robust value representations. Throughout, we use IB-based objective make representations robust."
>
>   (2) At the end of section 4, it would also help to list the objectives, or add the figure or pseudocode from the appendix. I understand you don't have much space, maybe decreasing the size of figures 1 and 2, or aggregating results over tasks could be a way to go here.
>
>   - **A1**. Thanks for your valuable feedback and suggestions regarding the structure of our paper.
>     - For (1), to improve clarity, we have revised the beginning of Section 4 to provide a clear "big picture" overview of the paper. We highlight the update in *blue* in **Lines 181-184** our revision, and for your convenience, we present it as follows.
>       - "In this paper, we propose to learn robust action-value representations that capture long-term information for decision-making. Then, we introduce robust and compressed reward representations and use TD learning paradigm to learn robust action-value representations, guided by an IB-based objective to ensure robustness throughout the learning process."
>     - For (2), we have moved the figure of ROUSER's architecture from the appendix to the end of Section 4 (see **Lines 378-396**) in our revision. Moreover, we have included the proposed objectives at the end of Section 4 (see **Lines 404-406**) to enhance readability. We highlight all these changes in *blue*.
>       - Due to the page limitation, we have merged the figures related to the DCS results from our main experiments into Figure 2 (Lines 432-454) and resized Figures 2, 3 (Lines 470-479), and 5 (Lines 499-509) in our revision. We thank you for the valuable suggestions to help streamline the content.
> - **Q2**.
>
>   (1) Line 506: What information are you predicting in this frozen evaluation? Is it an average of 300 rewards in the future?
>
>   (2) What exactly is the MSE between?
>
>   (3) Line 502: all methods' prediction: what does that mean?
>
>   (4) Line 504: "use a 2 layer mlp with a learning rate of 1e4". I understand what you're saying, but the sentence is awkward.
>   - **A2**. We apologize for any confusion caused by our unclear descriptions. We response to each comment as follows.
>     - For (1): No, we predict vectors with a dimension of 300, which are the future reward sequences over a length of 300.
>     - For (2): We compute the **MSE** between the **MLP outputs** and samples of **future reward sequences**.
>       - In this experiment, we first use a 2-layer trainable MLP to map the fixed representations into outputs, which are vectors with a dimension of 300. These fixed representations are extracted from the center layer of the critic in each method, using state-action pairs as inputs. We then compute the MSE between the MLP outputs and the samples of future reward sequences. A lower MSE indicates that the representations are better at capturing long-term information.
>     - For (3) and (4): Thank you for pointing this out. We have revised the whole paragraph as follows (also see **Lines 499-510** in our revision).
>       - "We assess how ROUSER can capture long-term robust information. Specifically, we first collect 200K samples under unseen color distractions using a DrQv2's policy. Next, we compute the representations learned by each method from state-action pairs in the collected data. Such representations are extracted from the center layer of the critic in each method. Then, we fix these representations and input them into a 2-layer trainable MLP to predict the future reward sequences over a length of 300. During the prediction process, we use the Adam optimizer with a learning rate of 1e-4. As shown in Figure 4(a), we plot curves of the MSE loss on 1K evaluation samples across three seeds. The lower MSE losses of ROUSER indicate that its learned representations capture long-term information more effectively."

---

> > ### Author Response · Authors · 2024-11-21
> >
> > - **Q3**. Line 227: "one of action-value representations" : unclear wording, what does "one of" mean here?
> >   - **A3**. We apologize for any confusion caused by our unclear wording. As derived in Appendix C.1, each action-value representation is linearly correlated with its infinite sequence of reward representations. Therefore, in our revision, we have removed "one of" from Line 228.
> > - **Q4**. Typos and unclear presentations.
> >
> >    (1) Line 270: we support that a compressed reward representation...
> >
> >    (2) Line 299: model a reward model -> introduce a reward model
> >
> >    (3) Line 364: supports -> allows
> >
> >    (4) Sections 5.2 and 5.3 titles are not very informative.
> >
> >    (5) Figure 4: please put the x and y axis labels in the figure itself, at least in a small font
> >
> >    (6) Line 466: Reword to something like "In this subsection, motivated by the result in theorem 4.4, we extend ROUSER to..."
> >
> >    (7) Line 515: by searchings alpha... : reword this, this is not grammatically correct
> >
> >    (8) Line 539: "sheds light" -- instead say "we believe this method is general and applicable to other problems, such as self-driving".
> >
> >   - **A4**. We deeply appreciate your time and effort. In our revision, we have corrected all these typos and revised unclear presentations, with these updates highlighted in *blue*. The details are as follows.
> >     - For (1), we replaced the typo "support" with "assume" in Lines 270 and 300 of our revision.
> >     - For (2), please refer to Line 299 in our revision.
> >     - For (3), please refer to Line 364 in our revision.
> >     - For (4), we updated the section titles:
> >       - i) changed the title of Section 5.1 to "Generalization Performance on DMC Benchmark with Continuous Actions" (see Line 418);
> >       - ii) changed the title of Section 5.2 to "Extending ROUSER to Procgen with Discrete Actions" (see Line 468);
> >       - iii) changed the title of Section 5.3 to "Analyzing the Representations learned by ROUSER" (see Line 497).
> >     - For (5), we updated each subplot in Figure 4 to include x- and y-axis labels directly on the plots (see Lines 486-495), instead of referencing them in the caption.
> >     - For (6), we revised the sentence: "In this subsection, motivated by the result in Theorem 4.4, we extend ROUSER to decision tasks with discrete actions, assessing its applicability." Please refer to Lines 470-472.
> >     - For (7), please refer to Lines 515-516. The update is: "Moreover, by tuning $\alpha$ in the conditional IB loss from Equation 11, we find that larger values of $\alpha$ result in smaller $\mathcal{D_{SKL}}$, potentially improving generalization performance."
> >     - For (8), please refer to Lines 538-539 in our revision.
> >   - Moreover, we have revised Lines 420-421, 424, 428, 511-512, and 518 to enhance readability. If you have any further comments, please let us know, and we will continue actively responding to them, improving our submissions.

---

> > > ### Author Response · Authors · 2024-11-23
> > > **Eagerly await your valuable feedback.**
> > >
> > > Dear Reviewer uw2P,
> > >
> > > We deeply appreciate your valuable feedback and the time you've taken to review our work, especially during this busy period.
> > >
> > > We are reaching out to kindly inquire about the current status of your review regarding our submission. We sincerely hope that our responses have adequately addressed your concerns. Furthermore, we are eager to address any additional queries you might have, which will enable us to enhance our work further.
> > >
> > > Once again, thank you for your guidance and support.
> > >
> > > Best,
> > > Authors

---

> > > > ### Comment · Reviewer_uw2P · 2024-11-25
> > > >
> > > > Thank you for the detailed response and for the changes to the manuscript!
> > > > You addressed my questions and concerns. I think this is a good paper, it presents a nice idea and tests it thoroughly, showing improved performance. I maintain my score unchanged (8).

---

> > > > > ### Author Response · Authors · 2024-11-25
> > > > >
> > > > > Dear Reviewer uw2P,
> > > > >
> > > > > Thanks for your kind support and for helping us improve the paper! We are deeply grateful for your thoughtful comments and for the confidence you have shown in our work.
> > > > >
> > > > > Thanks in advance!
> > > > >
> > > > > Best,
> > > > >
> > > > > Authors

---

### Official Review · Reviewer_238T · 2024-11-04

**Soundness:** 3
**Presentation:** 3
**Contribution:** 2
**Rating:** 5
**Confidence:** 4

**Summary:**

This paper proposes robust action-value representation learning (ROUSER). ROUSER aims to acquire robust representation by encouraging the representation to capture Q-values while minimizing mutual information with state-action pairs to discard irrelevant features. ROUSER is applicable to different VRL algorithms and can improve performance in environments with visual distractions.

**Strengths:**

- ROUSER aims to keep essential information by predicting the Q-value while neglecting irrelevant features by minimizing mutual information with state-action pairs.
- ROUSER's objective is applicable to common VRL algorithms.

**Weaknesses:**

- The idea of incorporating Q-values to mitigate visual distractions is a commonly adopted design choice in VRL, e.g. [1,4]. I believe the effectiveness of ROUSER needs to be evaluated in a more standard setting with VRL generalization baselines.
- To be specific:
    - The [DMC-GB](https://github.com/nicklashansen/dmcontrol-generalization-benchmark) is a commonly adopted visual generalization benchmark with `color_hard` and `video_hard` setting [1,2]. I believe the evaluation results on DMC-GB would be more persuasive.
    - For baselines, I believe comparing with generalization baselines is necessary. For example:
        - TPC [3] proposes to regularize the representation to encode temporally predictable features only.
        - [4] also proposes an information-bottleneck perspective to regularize representation.
        - TIA [5] designs structured representation so that the task-relevant representation captures long-term rewards while the task-irrelevant one captures no reward information.
        - PIEG [6] shows that representation from a pre-trained image encoder can boost the visual generalization significantly.


[1] Look where you look! Saliency-guided Q-networks for generalization in visual Reinforcement Learning

[2] Stabilizing Deep Q-Learning with ConvNets and Vision Transformers under Data Augmentation

[3] Temporal Predictive Coding For Model-Based Planning In Latent Space

[4] How to Learn Domain-Invariant Representations for Visual Reinforcement Learning - An Information-Theoretical Perspective

[5] Learning Task Informed Abstractions

[6] Pre-Trained Image Encoder for Generalizable Visual Reinforcement Learning

**Questions:**

NA

---

> ### Author Response · Authors · 2024-11-21
>
> We appreciate the insightful and constructive comments. We respond to each comment as follows and accordingly revise our manuscript, with all updates highlighted in **blue** for your convenience. We sincerely hope that our responses could properly address your concerns. If so, we would deeply appreciate it if you could raise your score. If not, please let us know your further concerns, and we will continue actively responding to the comments and enhancing our submission.
>
> - **Q1**. The idea of incorporating Q-values to mitigate visual distractions is a commonly adopted design choice in VRL, e.g. [1,2].
>   - **A1**. Thank you for your valuable comments. We present the major contributions of ROUSER as follows.
>     1. ROUSER provides a theoretical guarantee, establishing an **upper bound** between the true action-value function and the action-value function on top of learned representations (see Theorem 4.4 in our main text). In contrast, [1,2] only derive their optimization objectives without offering any guarantees.
>     2. We integrate ROUSER with both policy gradient and value-based VRL algorithms to estimate the action values, making it **suitable** for **continuous** and **discrete** control tasks. In contrast, [1,2] have only been implemented and evaluated on continuous control tasks.
>     3. To the best of our knowledge, ROUSER is the first to **extend** action-value representations to VRL against visual distractions, providing a new perspective on task decomposition and the development of VRL methods.
>     4. We have conducted additional experiments on **six DMC-GB tasks** across **all four settings** to comprehensively demonstrate that ROUSER outperforms the generalization baselines (including [1,2]) in **17 out of 24** settings. Please refer to A2 below for details.
>
>   [1] Look where you look! Saliency-guided Q-networks for generalization in visual Reinforcement Learning. NeurIPS 2022.
>
>   [2] How to Learn Domain-Invariant Representations for Visual Reinforcement Learning - An Information-Theoretical Perspective. IJCAI 2024.
>
> - **Q2**. (1) The DMC-GB is a commonly adopted visual generalization benchmark with color_hard and video_hard setting [1,2]. I believe the evaluation results on DMC-GB would be more persuasive. (2) For baselines, I believe comparing with generalization baselines is necessary. For example: TPC [3], MIIR [4], TIA [5], and PIEG [6].
>   - **A2**. Thank you for the valuable feedback.
>     - For (1), to comprehensively evaluate our approach, we have conducted additional experiments on **six DMC-GB tasks** across **all four settings** (i.e., `color_easy`, `color_hard`, `video_easy`, and `video_hard`). Please refer to *Tables B1, B2, B3, and B4* below.
>     - For (2), we have incorporated results from **six** VRL generalization baselines (including **DrG** [7], **MIIR**, **PIE-G**, SVEA [8], SGQN [9], and SECANT [10]) into our evaluations (also see Tables B1, B2, B3, and B4).
>       - Note that we replaced TPC with DrG, a model-based VRL generalization method similar to TPC, which uses the current state space model to enhance the robustness of latent representations against visual distractions. DrG demonstrates better generalization performance on DMC-GB than TPC.
>       - Due to computational resource constraints, the TIA experiments are still ongoing. We are committed to reporting TIA's results on DMC-GB as soon as they are completed.
>
>       All tables show that ROUSER outperforms the aforementioned baselines in **17 out of 24** settings, demonstrating ROUSER's robustness and effectiveness in DMC-GB tasks.
>
>   [1] Look where you look! Saliency-guided Q-networks for generalization in visual Reinforcement Learning. NeurIPS 2022.
>
>   [2] Stabilizing Deep Q-Learning with ConvNets and Vision Transformers under Data Augmentation. NeurIPS 2021.
>
>   [3] Temporal Predictive Coding For Model-Based Planning In Latent Space. ICML 2021.
>
>   [4] How to Learn Domain-Invariant Representations for Visual Reinforcement Learning - An Information-Theoretical Perspective. IJCAI 2024.
>
>   [5] Learning Task Informed Abstractions. ICML 2021.
>
>   [6] Pre-Trained Image Encoder for Generalizable Visual Reinforcement Learning. ICML 2022.
>
>   [7] Dream to Generalize: Zero-Shot Model-Based Reinforcement Learning for Unseen Visual Distractions. AAAI 2023.
>
>   [8] Stabilizing Deep Q-Learning with ConvNets and Vision Transformers under Data Augmentation. NeurIPS 2021.
>
>   [9] Look where you look! Saliency-guided Q-networks for generalization in visual Reinforcement Learning. NeurIPS 2022.
>
>   [10] Secant: Self-expert cloning for zero-shot generalization of visual policies. ICML 2021.

---

> ### Author Response · Authors · 2024-11-21
> **Tables B1 and B2**
>
> - We conducted comprehensive experiments across all four DMC-GB settings: `color_easy`, `color_hard`, `video_easy`, and `video_hard`.
>   - Most of the baseline results we report are taken directly from their respective papers. For cases where results are not provided, we ran experiments using the available source code and hyperparameters under three random seeds.
>     - We ran PIE-G, DrG, and SGQN on all DMC-GB tasks across `color_easy` and `color_hard` settings.
>     - We ran DrG on finger_spin and walker_stand tasks across `video_easy` and `video_hard` settings.
>     - We ran SVEA on all DMC-GB tasks under the `color_easy` setting and on cheetah_run task under the `color_hard` setting.
>   - Note that since SECANT and MIIR did not release their source code, we only included the results reported in their respective papers.
> - We have included all these updates in Appendix E.4 of our revision, including Tables B1 and B2 (see Table 7 in Lines 1422–1436), Tables B3 and B4 (see Table 8 in Lines 1467–1480), and their descriptions (see Lines 1394-1403). For your convenience, we have highlighted these updates in blue.
>
> ---
> **Table B1**. Results and standard deviations on DMC-GB under the `color_easy` setting. Results of our approach are over three random seeds.
> |DMC-GB Task| ROUSER           | PIE-G        | DrG          | SVEA         | SGQN         |
> | ----------------- | ---------------- | ------------ | ------------ | ------------ | ------------ |
> | ball_in_cup_catch | **963 $\pm$ 8**  | 955 $\pm$ 4  | 831 $\pm$ 92 | 959 $\pm$ 2  | 907 $\pm$ 71 |
> | cartpole_swingup  | **841 $\pm$ 20** | 624 $\pm$ 52 | 701 $\pm$ 43 | 826 $\pm$ 20 | 598 $\pm$ 92 |
> | cheetah_run       | **616 $\pm$ 13** | 429 $\pm$ 12 | 375 $\pm$ 31 | 587 $\pm$ 39 | 304 $\pm$ 43 |
> | finger_spin       | **933 $\pm$ 11** | 845 $\pm$ 5  | 876 $\pm$ 79 | 892 $\pm$ 59 | 628 $\pm$ 46 |
> | walker_walk       | **942 $\pm$ 6**  | 909 $\pm$ 40 | 812 $\pm$ 33 | 907 $\pm$ 23 | 845 $\pm$ 26 |
> | walker_stand      | **971 $\pm$ 2**  | 968 $\pm$ 4  | 910 $\pm$ 20 | 965 $\pm$ 10 | 963 $\pm$ 11 |
>
> ---
> **Table B2**. Results and standard deviations on DMC-GB under the `color_hard` setting. Results of our approach are over three random seeds.
> |DMC-GB Task| ROUSER          | PIE-G        | DrG           | SVEA          | SGQN          | SECANT           |
> | ----------------- | --------------- | ------------ | ------------- | ------------- | ------------- | ---------------- |
> | ball_in_cup_catch | **964 $\pm$ 5** | 960 $\pm$ 3  | 607 $\pm$ 46  | 961 $\pm$ 7   | 905 $\pm$ 71  | 958 $\pm$ 7      |
> | cartpole_swingup  | 799 $\pm$ 24    | 520 $\pm$ 69 | 523 $\pm$ 38  | 837 $\pm$ 23  | 540 $\pm$ 76  | **866 $\pm$ 15** |
> | cheetah_run       | 562 $\pm$ 10    | 376 $\pm$ 27 | 219 $\pm$ 8   | 456 $\pm$ 62  | 277 $\pm$ 43  | **582 $\pm$ 64** |
> | finger_spin       | **979 $\pm$ 9** | 838 $\pm$ 10 | 758 $\pm$ 124 | 977 $\pm$ 5   | 461 $\pm$ 5   | 910 $\pm$ 115    |
> | walker_walk       | **916 $\pm$ 9** | 824 $\pm$ 92 | 725 $\pm$ 134 | 760 $\pm$ 145 | 692 $\pm$ 153 | 856 $\pm$ 31     |
> | walker_stand      | **962 $\pm$ 2** | 948 $\pm$ 15 | 731 $\pm$ 21  | 942 $\pm$ 26  | 905 $\pm$ 34  | 939 $\pm$ 7      |

---

> ### Author Response · Authors · 2024-11-21
> **Tables B3 and B4**
>
> **Table B3**. Results and standard deviations on DMC-GB under the `video_easy` setting. Results of our approach are over three random seeds.
> |DMC-GB Task| ROUSER           | PIE-G         | DrG          | SVEA         | SGQN         | MIIR             | SECANT           |
> | ----------------- | ---------------- | ------------- | ------------ | ------------ | ------------ | ---------------- | ---------------- |
> | ball_in_cup_catch | **979 $\pm$ 21** | 922 $\pm$ 20  | 701 $\pm$ 36 | 871 $\pm$ 22 | 950 $\pm$ 24 | 973 $\pm$ 2      | 903 $\pm$ 49     |
> | cartpole_swingup  | 830 $\pm$ 21     | 587 $\pm$ 61  | 572 $\pm$ 25 | 782 $\pm$ 27 | 761 $\pm$ 28 | **858 $\pm$ 16** | 752 $\pm$ 38     |
> | cheetah_run       | 397 $\pm$ 30     | 287 $\pm$ 20  | 547 $\pm$ 21 | 249 $\pm$ 20 | 308 $\pm$ 34 | 393 $\pm$ 57     | **428 $\pm$ 70** |
> | finger_spin       | 970 $\pm$ 6      | 837 $\pm$ 107 | 751 $\pm$ 43 | 808 $\pm$ 33 | 956 $\pm$ 26 | **978 $\pm$ 9**  | 861 $\pm$ 102    |
> | walker_walk       | **925 $\pm$ 17** | 871 $\pm$ 22  | 902 $\pm$ 23 | 819 $\pm$ 71 | 910 $\pm$ 24 | 919 $\pm$ 30     | 842 $\pm$ 47     |
> | walker_stand      | **973 $\pm$ 3**  | 957 $\pm$ 12  | 910 $\pm$ 17 | 961 $\pm$ 8  | 955 $\pm$ 9  | 971 $\pm$ 3      | 932 $\pm$ 15     |
>
> ---
> **Table B4**. Results and standard deviations on DMC-GB under the `video_hard` setting. Results of our approach are over three random seeds.
> |DMC-GB Task| ROUSER           | PIE-G        | DrG              | SVEA          | SGQN         | MIIR             |
> | ----------------- | ---------------- | ------------ | ---------------- | ------------- | ------------ | ---------------- |
> | ball_in_cup_catch | **940 $\pm$ 5**  | 786 $\pm$ 47 | 635 $\pm$ 26     | 403 $\pm$ 174 | 782 $\pm$ 57 | 929 $\pm$ 9      |
> | cartpole_swingup  | **769 $\pm$ 37** | 401 $\pm$ 21 | 545 $\pm$ 23     | 393 $\pm$ 45  | 544 $\pm$ 43 | 765 $\pm$ 22     |
> | cheetah_run       | 270 $\pm$ 12     | 154 $\pm$ 17 | **489 $\pm$ 11** | 105 $\pm$ 37  | 135 $\pm$ 44 | 268 $\pm$ 73     |
> | finger_spin       | 937 $\pm$ 10     | 762 $\pm$ 59 | 437 $\pm$ 61     | 335 $\pm$ 58  | 822 $\pm$ 24 | **956 $\pm$ 18** |
> | walker_walk       | **868 $\pm$ 59** | 600 $\pm$ 28 | 782 $\pm$ 37     | 377 $\pm$ 93  | 739 $\pm$ 21 | 821 $\pm$ 58     |
> | walker_stand      | **968 $\pm$ 4**  | 852 $\pm$ 56 | 819 $\pm$ 58     | 834 $\pm$ 46  | 851 $\pm$ 24 | 965 $\pm$ 2      |

---

> > ### Author Response · Authors · 2024-11-23
> > **Eagerly await your valuable feedback.**
> >
> > Dear Reviewer 238T,
> >
> > We deeply appreciate your valuable feedback and the time you've taken to review our work, especially during this busy period.
> >
> > We are reaching out to kindly inquire about the current status of your review regarding our submission. We sincerely hope that our responses have adequately addressed your concerns. Furthermore, we are eager to address any additional queries you might have, which will enable us to enhance our work further.
> >
> > Once again, thank you for your guidance and support.
> >
> > Best,
> > Authors

---

> ### Author Response · Authors · 2024-11-27
> **We report all DMC-GB Results, including TIA.**
>
> Dear Reviewer 238T,
>
> We deeply appreciate the time and effort you've invested in reviewing our work.
>
> We have completed the additional experiments of TIA on six DMC-GB tasks across all four settings and updated all DMC-GB results in **Tables D1 and D2** below. In addition, we have updated **Appendix E.4 in our revision**, where Table D1 is presented as Table 7 in Lines 1422–1436, and Table D2 is presented as Table 8 in Lines 1459–1472.
> - Since the TIA agent based on its original source code always produced NaN outputs, we used the reproduced version of TIA provided at https://github.com/zchuning/repo.
>
> The results remain unchanged, demonstrating that ROUSER outperforms the aforementioned baselines in **17 out of 24** DMC-GB experiments.
>
> We hope our responses have thoroughly addressed your concerns, and we remain fully prepared to address any additional questions you may have to further refine our manuscript. If you find the revisions satisfactory, we would be grateful if you could **raise your score**.
>
> Thank you once again for your invaluable guidance and support.
>
> Best,
>
> Authors

---

> > ### Author Response · Authors · 2024-11-27
> > **Table D1**
> >
> > **Table D1**. Results and standard deviations on DMC-GB under the `color_easy` and `color_hard` settings. Results of our approach are averaged over three random seeds.
> >
> > | Task                   | **ROUSER**      | **PIE-G**          | **DrG**            | **SVEA**           | **SQGN**           | **TIA**            | **SECANT**         |
> > |------------------------|---------------------|---------------------|---------------------|---------------------|---------------------|---------------------|---------------------|
> > | **color_easy**         |                     |                     |                     |                     |                     |                     |                     |
> > | ball_in_cup_catch      | **963 ± 8**         | 955 ± 4            | 831 ± 92           | 959 ± 2            | 907 ± 71           | 652 ± 402          | -                  |
> > | cartpole_swingup       | **841 ± 20**        | 624 ± 52           | 701 ± 43           | 826 ± 20           | 598 ± 92           | 483 ± 62           | -                  |
> > | cheetah_run            | **616 ± 13**        | 429 ± 12           | 375 ± 31           | 587 ± 39           | 304 ± 43           | 281 ± 2            | -                  |
> > | finger_spin            | **933 ± 11**        | 845 ± 5            | 876 ± 79           | 892 ± 59           | 628 ± 46           | 410 ± 49           | -                  |
> > | walker_walk            | **942 ± 6**         | 909 ± 40           | 812 ± 33           | 907 ± 23           | 845 ± 26           | 780 ± 140          | -                  |
> > | walker_stand           | **971 ± 2**         | 968 ± 4            | 910 ± 20           | 965 ± 10           | 963 ± 11           | 895 ± 39           | -                  |
> > | **color_hard**         |                     |                     |                     |                     |                     |                     |                     |
> > | ball_in_cup_catch      | **964 ± 5**         | 960 ± 3            | 607 ± 46           | 961 ± 7            | 905 ± 71           | 575 ± 375          | 958 ± 7            |
> > | cartpole_swingup       | 799 ± 24            | 520 ± 69           | 523 ± 38           | 837 ± 23           | 540 ± 76           | 407 ± 173          | **866 ± 15**        |
> > | cheetah_run            | 562 ± 10            | 376 ± 27           | 219 ± 8            | 456 ± 62           | 277 ± 43           | 249 ± 25           | **582 ± 64**        |
> > | finger_spin            | **979 ± 9**         | 838 ± 10           | 758 ± 124          | 977 ± 5            | 461 ± 5            | 408 ± 79           | 910 ± 115          |
> > | walker_walk            | **916 ± 9**         | 824 ± 92           | 725 ± 134          | 760 ± 145          | 692 ± 153          | 616 ± 243          | 856 ± 31           |
> > | walker_stand           | **962 ± 2**         | 948 ± 15           | 731 ± 21           | 942 ± 26           | 905 ± 34           | 841 ± 37           | 939 ± 7            |

---

> > ### Author Response · Authors · 2024-11-27
> > **Table D2**
> >
> > **Table D2**. Results and standard deviations on DMC-GB under the `video_easy` and `video_hard` settings. Results of our approach are averaged over three random seeds.
> >
> > | Task                   | **ROUSER**         | **PIE-G**          | **DrG**            | **SVEA**           | **SQGN**           | **MIIR**           | **TIA**            | **SECANT**         |
> > |------------------------|--------------------|--------------------|--------------------|--------------------|--------------------|--------------------|--------------------|--------------------|
> > | **video_easy**         |                    |                    |                    |                    |                    |                    |                    |                    |
> > | ball_in_cup_catch      | **979 ± 21**       | 922 ± 20           | 701 ± 36           | 871 ± 22           | 950 ± 24           | 973 ± 2            | 664 ± 416          | 903 ± 49           |
> > | cartpole_swingup       | 830 ± 21           | 587 ± 61           | 572 ± 25           | 782 ± 27           | 761 ± 28           | **858 ± 16**       | 326 ± 139          | 752 ± 38           |
> > | cheetah_run            | 397 ± 30           | 287 ± 20           | 547 ± 21           | 249 ± 20           | 308 ± 34           | 393 ± 57           | 225 ± 41           | **428 ± 70**       |
> > | finger_spin            | 970 ± 6            | 837 ± 107          | 751 ± 43           | 808 ± 33           | 956 ± 26           | **978 ± 9**        | 295 ± 16           | 861 ± 102          |
> > | walker_walk            | **925 ± 17**       | 871 ± 22           | 902 ± 23           | 819 ± 71           | 910 ± 24           | 919 ± 30           | 696 ± 245          | 842 ± 47           |
> > | walker_stand           | **973 ± 3**        | 957 ± 12           | 910 ± 17           | 961 ± 8            | 955 ± 9            | 971 ± 3            | 903 ± 18           | 932 ± 15           |
> > | **video_hard**         |                    |                    |                    |                    |                    |                    |                    |                    |
> > | ball_in_cup_catch      | **940 ± 5**        | 786 ± 47           | 635 ± 26           | 403 ± 174          | 782 ± 57           | 929 ± 9            | 314 ± 36           | -                  |
> > | cartpole_swingup       | **769 ± 37**       | 401 ± 21           | 545 ± 23           | 393 ± 45           | 544 ± 43           | 765 ± 22           | 189 ± 29           | -                  |
> > | cheetah_run            | 270 ± 12           | 154 ± 17           | **489 ± 11**       | 105 ± 37           | 135 ± 44           | 268 ± 73           | 76 ± 25            | -                  |
> > | finger_spin            | 937 ± 10           | 762 ± 59           | 437 ± 61           | 335 ± 58           | 822 ± 24           | **956 ± 18**       | 182 ± 55           | -                  |
> > | walker_walk            | **868 ± 59**       | 600 ± 28           | 782 ± 37           | 377 ± 93           | 739 ± 21           | 821 ± 58           | 166 ± 60           | -                  |
> > | walker_stand           | **968 ± 4**        | 852 ± 56           | 819 ± 58           | 834 ± 46           | 851 ± 24           | 965 ± 2            | 673 ± 142          | -                  |

---

> > > ### Author Response · Authors · 2024-11-29
> > > **This is a kind Reminder.**
> > >
> > > Dear Reviewer 238T,
> > >
> > > We deeply appreciate the time and effort you've invested in reviewing our work.
> > >
> > > As the discussion deadline of **December 2nd** approaches, if convenient, could you please let us know your feedback on our submission? We hope our responses have thoroughly addressed your concerns, and we remain fully prepared to address any additional questions you may have to further refine our manuscript. If you find the revisions *satisfactory*, we would be grateful if you could *raise your score*.
> > >
> > > Thank you once again for your invaluable guidance and support.
> > >
> > > Best,
> > >
> > > Authors

---

> > > > ### Author Response · Authors · 2024-12-02
> > > > **A Gentle Reminder.**
> > > >
> > > > Dear Reviewer 238T,
> > > >
> > > > We would like to express our sincere gratitude once again for your feedback and constructive suggestions. Your guidance has been invaluable in helping us improve the quality of our work!
> > > >
> > > > We are writing to gently remind you that **the author-reviewer discussion period will end in less than 24 hours**. We eagerly await your feedback **to understand if our responses have adequately addressed your concerns**. **If so, we would deeply appreciate it if you could raise your score**. If not, we are eager to address any queries you might have, which will enable us to further enhance our work.
> > > >
> > > > Best,
> > > >
> > > > Authors

---

> > > > > ### Author Response · Authors · 2024-12-03
> > > > > **A Gentle Reminder**
> > > > >
> > > > > Dear Reviewer 238T,
> > > > >
> > > > > We sincerely appreciate the time and effort you have dedicated to reviewing our paper.
> > > > >
> > > > > As a gentle reminder, the discussion period will conclude in **5 hours**. We fully understand that you may have other pressing commitments or a particularly busy schedule. In light of this, and to assist in **saving your valuable time**, we have carefully summarized the key points of our rebuttal as follows.
> > > > >
> > > > > - In Appendix E.4 of our revision, (1) we have included comprehensive experiments on *six DMC-GB tasks* across *all four settings* (i.e., color_easy, color_hard, video_easy, and video_hard), and (2) we have incorporated DMC-GB results of *seven VRL generalization baselines*, including those specifically mentioned by you.
> > > > >   These results demonstrate that ROUSER outperforms the baselines in *17 out of 24* settings.
> > > > > - We explain that our primary contributions extend beyond the VRL generalization methods SGQN and MIIR.
> > > > >   1. Our approach ROUSER provides a theoretical guarantee, establishing an *upper bound* between the true action-value function and the action-value function on top of learned representations (see Theorem 4.4 in our main text). In contrast, SGQN and MIIR only derive their optimization objectives without offering any guarantees.
> > > > >   2. We integrate ROUSER with both policy gradient and value-based VRL algorithms to estimate the action values, making it *suitable for continuous and discrete control tasks*. In contrast, SGQN and MIIR have only been implemented and evaluated on continuous control tasks.
> > > > >   3. To the best of our knowledge, ROUSER is the first to *extend action-value representations to VRL against visual distractions*, providing a new perspective on task decomposition and the development of VRL methods.
> > > > >
> > > > > **We sincerely hope that our rebuttal can properly address your concerns. If so, we would appreciate it if you could consider raising your score**. If not, please let us know your remaining concerns or suggestions. We will do our utmost to address your further concerns and make any necessary improvements.
> > > > >
> > > > > Many thanks in advance for taking the time to read our rebuttal.
> > > > >
> > > > > Best regards,
> > > > >
> > > > > Authors

---

> > > > > > ### Author Response · Authors · 2024-12-03
> > > > > >
> > > > > > Dear Reviewer 238T,
> > > > > >
> > > > > > We are writing as the authors of the paper "Learning Robust Representations with Long-Term Information for Generalization in Visual Reinforcement Learning" (ID: 8907). We sincerely appreciate the time and effort you have devoted to reviewing our paper.
> > > > > >
> > > > > > While we have not yet received feedback from you during the discussion phase, we fully understand that you may be managing other pressing commitments or navigating a hectic schedule.
> > > > > >
> > > > > > - We **humbly believe** that you recognize **the value and significance of our work**, as reflected in your review, where kindly describe our work as "keep essential information ... while neglecting irrelevant features ..." and "is applicable to common VRL algorithms."
> > > > > >
> > > > > > - Moreover, **we humbly believe that our rebuttal has thoroughly addressed your concerns**. We have carefully revised our paper and provided comprehensive experiments to improve our submission. To assist in saving your valuable time, we have also summarized the key points of our rebuttal for your convenience.
> > > > > >
> > > > > > As the discussion period will conclude in **2 hours**, we remain hopeful and sincerely look forward to receiving your valuable feedback, should your schedule allow.
> > > > > >
> > > > > > Thank you again for your time, effort, and thoughtful consideration.
> > > > > >
> > > > > > Best,
> > > > > >
> > > > > > Authors

---

### Official Review · Reviewer_M8ko · 2024-11-04

**Soundness:** 3
**Presentation:** 3
**Contribution:** 3
**Rating:** 6
**Confidence:** 3

**Summary:**

This paper proposes a novel approach termed Robust Action-Value Representation Learning (ROUSER) within the Information Bottleneck (IB) framework. This paper evaluates ROUSER on VRL generalization benchmark and conducts various experiments to demonstrate the advances of ROUSER.

**Strengths:**

1.The method part of this paper has sufficient theoretical analysis and proof.

2.The method is evaluated on various experiments.

**Weaknesses:**

See questions.

**Questions:**

1.The six tasks in the DeepMind Control (DMC) suite, including "reacher_easy," are commonly used to assess the sample efficiency of VRL. However, the DMC-GB also includes tasks like "walker_stand" for evaluating generalization capabilities. To provide a more comprehensive evaluation, you should run additional experiments on "walker_stand" to better assess your method's generalization performance in alignment with this benchmark.

2.While CURL and DrQv2 have been evaluated on generalization benchmarks, they were not originally designed to address the specific challenges of generalization in VRL. To provide a more relevant comparison, you should include results for your method alongside established methods like SECANT [1] and PIE-G [2], which were specifically developed to enhance generalization in VRL.

[1] Linxi Fan, Guanzhi Wang, De-An Huang, Zhiding Yu, Li Fei-Fei, Yuke Zhu, and Animashree Anand kumar. Secant: Self-expert cloning for zero-shot generalization of visual policies. In International Conference on Machine Learning, 2021.

[2] Zhecheng Yuan, Zhengrong Xue, Bo Yuan, Xueqian Wang, Yi Wu, Yang Gao, and Huazhe Xu. Pre trained image encoder for generalizable visual reinforcement learning. In First Workshop on Pre training: Perspectives, Pitfalls, and Paths Forward at ICML 2022, 2022.

---

> ### Author Response · Authors · 2024-11-21
>
> We appreciate the reviewer's insightful and constructive comments and suggestions. We respond to each comment as follows and accordingly revise our manuscript, with all updates highlighted in **blue** for your convenience. We sincerely hope that our responses could properly address your concerns. If so, we would deeply appreciate it if you could raise your score. If not, please let us know your further concerns, and we will continue actively responding to the comments and enhancing our submission.
>
> - **Q1**. To provide a more comprehensive evaluation, you should run additional experiments on "walker_stand" to better assess your method's generalization performance in alignment with this benchmark.
>   - **A1**. Thanks for the kind and insightful comment.
>     - For a comprehensive evaluation, we have conducted additional experiments on **six DMC-GB tasks** (including the "walker_stand" task) across **all four settings** (`color_easy`, `color_hard`, `video_easy`, and `video_hard`).
>       We report the average results and standard deviations over three seeds. These results demonstrate that ROUSER outperforms others in **17 out of 24** experiments. Please refer to *Tables B1, B2, B3, and B4* below for details.
>     - Moreover, it is worth noting that Distracting Control Suite (DistractingCS)---which we use in our paper to evaluate generalization capabilities of agents---introduces visual distractions (i.e., random colors and video backgrounds) similar to those of the DMC-GB [1], but with **higher stochasticity** (see [DMC-GB](https://github.com/nicklashansen/dmcontrol-generalization-benchmark)).
>
>    [1] Generalization in Reinforcement Learning by Soft Data Augmentation. ICRA 2021.
>
> - **Q2**. To provide a more relevant comparison, you should include results for your method alongside established methods like SECANT and PIE-G, which were specifically developed to enhance generalization in VRL.
>   - **A2**. Thanks for your suggestion.
>     - We have included results from **six** VRL generalization methods (including **PIE-G** [1], **SECANT** [2], DrG [3], SVEA [4], SGQN [5], and MIIR [6]) in our evaluations. *Tables B1, B2, B3, and B4* present these results on DMC-GB, demonstrating the **superiority** of ROUSER for generalization.
>       - Note that SECANT did not provide DMC-GB results under `color_easy` and `video_hard` settings, nor did it release the source code. Thus, we only included the results in `color_hard` and `video_easy` settings reported in its paper.
>
>    [1] Pre-trained image encoder for generalizable visual reinforcement learning. ICML 2022.
>
>    [2] Secant: Self-expert cloning for zero-shot generalization of visual policies. ICML 2021.
>
>    [3] Dream to Generalize: Zero-Shot Model-Based Reinforcement Learning for Unseen Visual Distractions. AAAI 2023.
>
>    [4] Stabilizing Deep Q-Learning with ConvNets and Vision Transformers under Data Augmentation. NeurIPS 2021.
>
>    [5] Look where you look! Saliency-guided Q-networks for generalization in visual Reinforcement Learning. NeurIPS 2022.
>
>    [6] How to Learn Domain-Invariant Representations for Visual Reinforcement Learning - An Information-Theoretical Perspective. IJCAI 2024.

---

> ### Author Response · Authors · 2024-11-21
> **Tables B1 and B2**
>
> - We conducted comprehensive experiments across all four DMC-GB settings: `color_easy`, `color_hard`, `video_easy`, and `video_hard`.
>   - Most of the baseline results we report are taken directly from their respective papers. For cases where results are not provided, we ran experiments using the available source code and hyperparameters under three random seeds.
>     - We ran PIE-G, DrG, and SGQN on all DMC-GB tasks across `color_easy` and `color_hard` settings.
>     - We ran DrG on finger_spin and walker_stand tasks across `video_easy` and `video_hard` settings.
>     - We ran SVEA on all DMC-GB tasks under the `color_easy` setting and on cheetah_run task under the `color_hard` setting.
>   - Note that since SECANT and MIIR did not release their source code, we only included the results reported in their respective papers.
> - We have included all these updates in Appendix E.4 of our revision, including Tables B1 and B2 (see Table 7 in Lines 1422–1436), Tables B3 and B4 (see Table 8 in Lines 1467–1480), and their descriptions (see Lines 1394-1403). For your convenience, we have highlighted these updates in blue.
>
> ---
> **Table B1**. Results and standard deviations on DMC-GB under the `color_easy` setting. Results of our approach are over three random seeds.
> |DMC-GB Task| ROUSER           | PIE-G        | DrG          | SVEA         | SGQN         |
> | ----------------- | ---------------- | ------------ | ------------ | ------------ | ------------ |
> | ball_in_cup_catch | **963 $\pm$ 8**  | 955 $\pm$ 4  | 831 $\pm$ 92 | 959 $\pm$ 2  | 907 $\pm$ 71 |
> | cartpole_swingup  | **841 $\pm$ 20** | 624 $\pm$ 52 | 701 $\pm$ 43 | 826 $\pm$ 20 | 598 $\pm$ 92 |
> | cheetah_run       | **616 $\pm$ 13** | 429 $\pm$ 12 | 375 $\pm$ 31 | 587 $\pm$ 39 | 304 $\pm$ 43 |
> | finger_spin       | **933 $\pm$ 11** | 845 $\pm$ 5  | 876 $\pm$ 79 | 892 $\pm$ 59 | 628 $\pm$ 46 |
> | walker_walk       | **942 $\pm$ 6**  | 909 $\pm$ 40 | 812 $\pm$ 33 | 907 $\pm$ 23 | 845 $\pm$ 26 |
> | walker_stand      | **971 $\pm$ 2**  | 968 $\pm$ 4  | 910 $\pm$ 20 | 965 $\pm$ 10 | 963 $\pm$ 11 |
>
> ---
> **Table B2**. Results and standard deviations on DMC-GB under the `color_hard` setting. Results of our approach are over three random seeds.
> |DMC-GB Task| ROUSER          | PIE-G        | DrG           | SVEA          | SGQN          | SECANT           |
> | ----------------- | --------------- | ------------ | ------------- | ------------- | ------------- | ---------------- |
> | ball_in_cup_catch | **964 $\pm$ 5** | 960 $\pm$ 3  | 607 $\pm$ 46  | 961 $\pm$ 7   | 905 $\pm$ 71  | 958 $\pm$ 7      |
> | cartpole_swingup  | 799 $\pm$ 24    | 520 $\pm$ 69 | 523 $\pm$ 38  | 837 $\pm$ 23  | 540 $\pm$ 76  | **866 $\pm$ 15** |
> | cheetah_run       | 562 $\pm$ 10    | 376 $\pm$ 27 | 219 $\pm$ 8   | 456 $\pm$ 62  | 277 $\pm$ 43  | **582 $\pm$ 64** |
> | finger_spin       | **979 $\pm$ 9** | 838 $\pm$ 10 | 758 $\pm$ 124 | 977 $\pm$ 5   | 461 $\pm$ 5   | 910 $\pm$ 115    |
> | walker_walk       | **916 $\pm$ 9** | 824 $\pm$ 92 | 725 $\pm$ 134 | 760 $\pm$ 145 | 692 $\pm$ 153 | 856 $\pm$ 31     |
> | walker_stand      | **962 $\pm$ 2** | 948 $\pm$ 15 | 731 $\pm$ 21  | 942 $\pm$ 26  | 905 $\pm$ 34  | 939 $\pm$ 7      |

---

> ### Author Response · Authors · 2024-11-21
> **Tables B3 and B4**
>
> **Table B3**. Results and standard deviations on DMC-GB under the `video_easy` setting. Results of our approach are over three random seeds.
> |DMC-GB Task| ROUSER           | PIE-G         | DrG          | SVEA         | SGQN         | MIIR             | SECANT           |
> | ----------------- | ---------------- | ------------- | ------------ | ------------ | ------------ | ---------------- | ---------------- |
> | ball_in_cup_catch | **979 $\pm$ 21** | 922 $\pm$ 20  | 701 $\pm$ 36 | 871 $\pm$ 22 | 950 $\pm$ 24 | 973 $\pm$ 2      | 903 $\pm$ 49     |
> | cartpole_swingup  | 830 $\pm$ 21     | 587 $\pm$ 61  | 572 $\pm$ 25 | 782 $\pm$ 27 | 761 $\pm$ 28 | **858 $\pm$ 16** | 752 $\pm$ 38     |
> | cheetah_run       | 397 $\pm$ 30     | 287 $\pm$ 20  | 547 $\pm$ 21 | 249 $\pm$ 20 | 308 $\pm$ 34 | 393 $\pm$ 57     | **428 $\pm$ 70** |
> | finger_spin       | 970 $\pm$ 6      | 837 $\pm$ 107 | 751 $\pm$ 43 | 808 $\pm$ 33 | 956 $\pm$ 26 | **978 $\pm$ 9**  | 861 $\pm$ 102    |
> | walker_walk       | **925 $\pm$ 17** | 871 $\pm$ 22  | 902 $\pm$ 23 | 819 $\pm$ 71 | 910 $\pm$ 24 | 919 $\pm$ 30     | 842 $\pm$ 47     |
> | walker_stand      | **973 $\pm$ 3**  | 957 $\pm$ 12  | 910 $\pm$ 17 | 961 $\pm$ 8  | 955 $\pm$ 9  | 971 $\pm$ 3      | 932 $\pm$ 15     |
>
> ---
> **Table B4**. Results and standard deviations on DMC-GB under the `video_hard` setting. Results of our approach are over three random seeds.
> |DMC-GB Task| ROUSER           | PIE-G        | DrG              | SVEA          | SGQN         | MIIR             |
> | ----------------- | ---------------- | ------------ | ---------------- | ------------- | ------------ | ---------------- |
> | ball_in_cup_catch | **940 $\pm$ 5**  | 786 $\pm$ 47 | 635 $\pm$ 26     | 403 $\pm$ 174 | 782 $\pm$ 57 | 929 $\pm$ 9      |
> | cartpole_swingup  | **769 $\pm$ 37** | 401 $\pm$ 21 | 545 $\pm$ 23     | 393 $\pm$ 45  | 544 $\pm$ 43 | 765 $\pm$ 22     |
> | cheetah_run       | 270 $\pm$ 12     | 154 $\pm$ 17 | **489 $\pm$ 11** | 105 $\pm$ 37  | 135 $\pm$ 44 | 268 $\pm$ 73     |
> | finger_spin       | 937 $\pm$ 10     | 762 $\pm$ 59 | 437 $\pm$ 61     | 335 $\pm$ 58  | 822 $\pm$ 24 | **956 $\pm$ 18** |
> | walker_walk       | **868 $\pm$ 59** | 600 $\pm$ 28 | 782 $\pm$ 37     | 377 $\pm$ 93  | 739 $\pm$ 21 | 821 $\pm$ 58     |
> | walker_stand      | **968 $\pm$ 4**  | 852 $\pm$ 56 | 819 $\pm$ 58     | 834 $\pm$ 46  | 851 $\pm$ 24 | 965 $\pm$ 2      |

---

> > ### Author Response · Authors · 2024-11-23
> > **Eagerly await your valuable feedback.**
> >
> > Dear Reviewer M8ko,
> >
> > We deeply appreciate your valuable feedback and the time you've taken to review our work, especially during this busy period.
> >
> > We are reaching out to kindly inquire about the current status of your review regarding our submission. We sincerely hope that our responses have adequately addressed your concerns. Furthermore, we are eager to address any additional queries you might have, which will enable us to enhance our work further.
> >
> > Once again, thank you for your guidance and support.
> >
> > Best,
> > Authors

---

> > ### Comment · Reviewer_M8ko · 2024-11-25
> >
> > Thank you for your reply. My concerns have been addressed, and I will maintain my rating of 6.

---

> ### Author Response · Authors · 2024-11-25
>
> Dear Reviewer M8ko,
>
> Thanks for your kind support and the further improvements you have suggested for our manuscript. We sincerely appreciate your insightful feedback, which has significantly helped us in ensuring a more comprehensive evaluation of our approach.
>
> Thank you again for your valuable comments and guidance.
>
> Best,
>
> Authors

---

### Meta-Review · Area_Chair_dC4V · 2024-12-23

**Metareview:**

The paper presents a compelling contribution to visual reinforcement learning through the ROUSER method. The authors addressed reviewers' concerns by conducting extensive experiments across multiple benchmarks, demonstrating ROUSER's performance in 17 out of 24 experimental settings. The method's key strengths include theoretical guarantees, applicability to both continuous and discrete control tasks, and a novel approach to learning robust representations using an information bottleneck framework. Reviewers acknowledged the paper's contributions, with some maintaining high scores and expressing satisfaction with the authors' detailed responses and improvements. The careful revisions and thorough addressing of specific concerns strongly support the paper's acceptance as a valuable contribution to visual reinforcement learning.

**Additional Comments On Reviewer Discussion:**

Reviewers raised concerns about experimental validation, presentation, and methodological depth in the paper. The authors responded by conducting comprehensive DMC-GB experiments, incorporating multiple generalization baselines, improving paper structure, and providing additional theoretical comparisons. They systematically addressed feedback by expanding analysis, clarifying theoretical contributions, and demonstrating ROUSER's performance across various tasks. Most reviewers found these responses satisfactory, supporting the paper's acceptance as a significant contribution to visual reinforcement learning.

---

### Decision · Program_Chairs · 2025-01-22

Accept (Poster)